# Air Quality Arena: A Large-Scale Multi-Region Ground Monitoring Dataset and Benchmark for Air Quality Forecasting with Time-Series Foundation Models

**Rishi Bharadwaj** [1]  **Manik Gupta** [1]  **Pandarasamy Arjunan** [2]

## Abstract

Air pollution causes an estimated 7.9 million premature deaths annually, making accurate forecasting a critical public health priority. Machine learning is increasingly being applied to forecast air pollution levels, yet existing benchmarks remain narrow in both geographic scope and pollutant coverage, and fail to evaluate the latest generation of time series foundation models (TSFMs) on real world, large scale data. We present Air Quality Arena (AQA), a large scale multi-country and multi-pollutant dataset (AQA-Data) and benchmark (AQA-Bench) to address this gap. AQA covers 6 major pollutants over a three year period across 7 diverse countries and 4 continents, with more than 14,000 station-pollutant series, aiming to provide a comprehensive benchmark for air quality tasks. We benchmark this dataset across 11 leading time series foundation models and classical baselines to assess performance on short-term air quality forecasting. Our results demonstrate that TSFMs are effective zero-shot forecasters and consistently outperform classical baselines, with our top-performing model employing a cross-modal architecture that leverages a vision foundation model for time series forecasting. AQA is publicly released at `AirQualityArena.github.io`.

## 1. Introduction

Air pollution is now the leading environmental cause of premature deaths globally, responsible for an estimated 7.9 million deaths annually (Health Effects Institute, 2025), sur-

passing tobacco as a risk factor. The World Health Organization estimates that 99% of the global population is exposed to air that exceeds safe pollutant thresholds, underscoring the urgent need for accurate, large scale air quality monitoring and forecasting. Early forecasting of pollution events can enable timely public health interventions, yet accurate prediction remains difficult due to the complex interplay of meteorological conditions, emission sources, and local geography.

Traditional approaches based on numerical models and statistical methods have seen growing competition from deep learning models. More recently, Time Series Foundation Models (TSFMs), large models pretrained on diverse time series corpora that can forecast in a zero-shot setting without task specific training (Ansari et al., 2024; Das et al., 2024; Woo et al., 2024), have come into prominence. This is particularly appealing for air quality forecasting, where labelled historical data may be sparse, unreliable, or simply unavailable for newly deployed stations, and where retraining a model for each new site or pollutant is impractical at scale. Despite this rapid progress in model development, the benchmarks used to evaluate air quality forecasting have not kept pace. Most studies evaluate a single country or region (Sharma & Mauzerall, 2022; Silver et al., 2025), focus on one or two pollutants, typically $PM_{2.5}$ (Su et al., 2023; Rakholia et al., 2022), and none systematically compare the new generation of TSFMs against classical baselines at scale across diverse geographies and multiple pollutants. We introduce AQA to address this gap.

Rather than relying on aggregated sources such as OpenAQ (OpenAQ), which we found failed minimum continuity requirements at almost every station evaluated, we collected raw measurements directly from six official national monitoring networks spanning the United States, India, China, the United Kingdom, Mexico, and France and Germany. These networks span four continents and 7 countries to capture climatic and emission diversity absent from prior benchmarks, from coal dominated industry in China to agricultural burning in India.

The three year time window (July 2022–June 2025) was chosen to be as recent as possible given the semi-annual update frequency of some sources, and to capture the seasonal vari-

---

[1]Department of Computer Science and Information Systems, BITS Pilani, Hyderabad Campus, Hyderabad, India [2]Department of Cyber-Physical Systems, Indian Institute of Science, Bengaluru, India. Correspondence to: Pandarasamy Arjunan <samy@iisc.ac.in>.

*Proceedings of the $2^{nd}$ ICML Workshop on Foundation Models for Structured Data*, Seoul, South Korea. 2026. Copyright 2026 by the author(s).

ability essential for pollutants such as $O_3$ and $PM_{2.5}$. AQA-Data covers 6 major pollutants ($PM_{2.5}$, $PM_{10}$, $NO_2$, $SO_2$, CO, and $O_3$) comprising more than 14,000 station-pollutant series. We benchmark this dataset across 11 leading TSFMs and 6 classical baselines for short-term air quality forecasting.

## 1.1. Contributions

We present AQA, which, to the best of our knowledge, is the largest and most geographically diverse ML ready air quality dataset and benchmark currently available. It spans regulatory monitoring networks across seven countries, six pollutants, and supports several tasks including forecasting, classification, and transfer learning. Below are our main contributions:

- **Dataset:** AQA-Data, an air quality dataset spanning 6 monitoring networks, 7 countries, 4 continents, and 6 pollutants over a three-year period (July 2022-June 2025).

- **Benchmark:** AQA-Bench, a standardised forecasting benchmark evaluated across 11 time series foundation models and 6 classical baselines for short-term air quality prediction.

- **Analysis:** A cross-geography and cross-pollutant evaluation revealing where TSFMs generalise as reliable zero-shot forecasters and where meaningful performance gaps remain, with implications for future model development and benchmark design.

- **Open codebase:** An extensible framework supporting easy integration of additional air quality networks, pollutants and models.

## 2. Related Work

### 2.1. Air Quality Datasets

Several publicly available air quality datasets have been developed to support environmental monitoring and machine learning research. However, each comes with notable limitations in terms of temporal coverage, pollutant diversity, geographic breadth, and suitability for time series forecasting.

**OpenAQ** aggregates ground-based measurements from government and research networks across more than 100 countries, accessible via a public API (OpenAQ). Its geographic breadth and standardized formatting are appealing, but data quality and continuity vary widely across stations. It performs no systematic gap filling, leaving preprocessing decisions to individual researchers. This inconsistency poses a significant obstacle for reproducible benchmarking. In our

evaluation, OpenAQ data failed to meet minimum continuity requirements at almost every station we assessed, despite drawing from many of the same underlying networks as AQA-Data. For example, based on our analysis, we found data from India with year long gaps. Data is also no longer being collected from China.

**AQICN** is a similarly broad aggregator, consolidating real time air quality data from thousands of stations worldwide (World Air Quality Index Project). However, AQICN currently exposes only calculated AQI indices rather than individual pollutant concentrations, making it unsuitable for multi-pollutant forecasting research.

**AQ-Bench** (Betancourt et al., 2021) covers over 5,500 stations globally, but is restricted to only ozone and does not include any other pollutants. Its annual aggregated resolution renders it inadequate for time series forecasting tasks.

A consistent pattern emerges from existing datasets. The few that cover multiple pollutants are limited to a single city or country. Those with global reach provide only aggregated statistics or composite indices, with no explicit data preprocessing for building ML pipelines. None address the geographic diversity, pollutant coverage, and pretraining considerations that rigorous TSFM benchmarking demands. AQA is proposed and designed to close all of these gaps simultaneously in a systematic manner.

### 2.2. Benchmarking Time Series Models

Seasonal Naive (Hyndman & Athanasopoulos, 2018) and AutoETS (Hyndman et al., 2008) represent the statistical baselines, simple but competitive on data with strong seasonal structure. LightGBM (Ke et al., 2017) is a gradient-boosted tree model trained on lag and calendar features, offering a strong non-neural supervised baseline. DeepAR (Salinas et al., 2020), DLinear (Zeng et al., 2023) and PatchTST (Nie et al., 2023) are supervised deep learning baselines. DeepAR uses an autoregressive recurrent neural network model to produce accurate probabilistic forecasts. DLinear uses simple linear models and has proven to be surprisingly competitive against complex Transformer architectures on standard benchmarks. PatchTST uses a patch-based Transformer with channel independence, achieving strong results on traffic, weather and electricity forecasting. Together, these baselines span classical statistics to top performing deep learning methods, providing a comprehensive reference to evaluate foundation model performance.

TSFMs such as Chronos, Moirai, and TimesFM (Ansari et al., 2024; Woo et al., 2024; Das et al., 2024) are pretrained on large, diverse time series corpora and forecast without any task-specific training. The closest existing work using TSFMs for air quality forecasting is Saurav et al. (2025), who evaluate TSFMs on atmospheric $CO_2$ forecasting under

zero-shot and fine-tuned settings and assess spatial transfer capabilities across locations. AQA extends this from a single atmospheric variable to six pollutants across seven countries. Crucially, our comparison is asymmetric by design, as TSFMs are evaluated zero-shot while supervised baselines are fitted per pollutant per network. This directly tests a practically relevant question, i.e, can a pretrained foundation model, without seeing any air quality data, compete with a model optimised on the target series?

## 3. AQA-Data

### 3.1. Data Sources and Collection

AQA-Data aggregates ground station measurements from six official air quality monitoring networks across seven countries. Environmental Protection Agency (EPA) hourly data was downloaded directly from the Air Quality System (AQS) public data repository (U.S. Environmental Protection Agency). Central Pollution Control Board (CPCB) Observations were collected from the Indian government's Continuous Ambient Air Quality Monitoring (CAAQM) portal. Automatic Urban and Rural Network (AURN) data for the United Kingdom (Department for Environment, Food & Rural Affairs) was retrieved using PyAURN (Wilson), a Python wrapper around the openair R package (Carslaw & Ropkins, 2012). Chinese national monitoring data was collected from quotsoft.net, a third-party aggregator of China National Environmental Monitoring Centre (CNEMC) station readings. European Environment Agency (EEA) data covering France and Germany was downloaded directly from the EEA's official data portal. Mexican data was collected from the Sistema Nacional de Información de la Calidad del Aire (SINAICA) government website (Gobierno de México).

To assess potential pretraining contamination, we inspected the publicly available corpora of the evaluated TSFMs. Overlap with AQA-Data was identified only at a small subset of CNEMC stations and AURN stations in London, with the remaining five networks entirely unaffected. For the overlapping stations, we found no temporal intersection between the pretraining data and AQA-Data. Given the limited scale of the overlap and the absence of temporal intersection, we consider contamination effects on the reported results negligible.

### 3.2. Preprocessing Framework

All collected data was converted into a uniform format organised by monitoring site and pollutant, covering three years of hourly readings from July 1 2022 to June 30 2025. To balance dataset quantity and quality, we restricted our analysis to monitoring sites that contained at least 70% valid observations and had no temporal gaps longer than

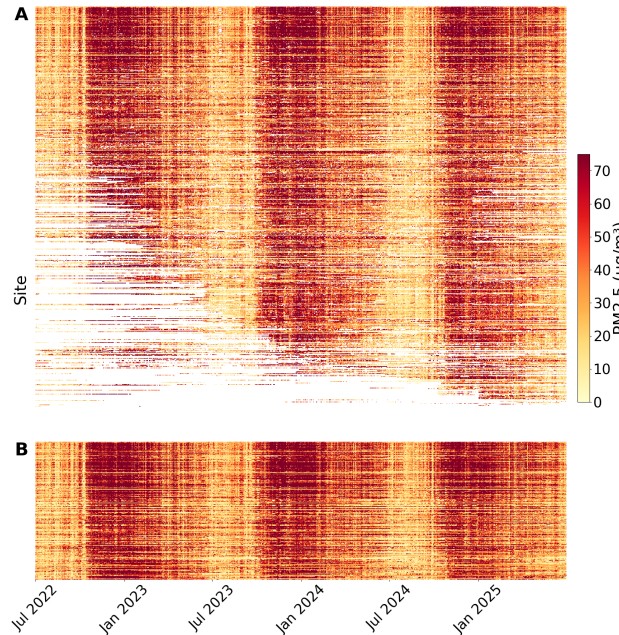

*Figure 1.* CPCB $PM_{2.5}$ heatmaps across monitoring sites over time, (A) before and (B) after transformation. White regions indicate missing observations. The completeness after transformation reflects removal of sites during filtering and MSTL-based gap imputation.

two weeks. This filtering was performed separately for each pollutant. Remaining gaps after site filtering were imputed using Multiple Seasonal-Trend decomposition using LOESS (MSTL) (Cleveland et al., 1990). Interpolating in the deseasonalized space produces more accurate gap estimates than naive linear interpolation for data with strong seasonality (Chhabra, 2023; Wijesekara & Liyanage, 2023). Imputed values were clipped to $[0, x_{max}]$, where $x_{max}$ is the maximum observed value for that site and pollutant, to prevent the introduction of artificial outliers. Figure 1 shows data from the CPCB network before and after transformation.

## 4. Benchmark Design

We adapt the TIME framework (Qiao et al., 2026) as our evaluation harness, extending it with additional models and training capabilities for our supervised baselines. All models use a 168-hour context window and forecast the next 24 hours, evaluated on a rolling window with a step size of 24. All models operate in a strictly univariate setting with no covariates. Due to the large number of stations in CNEMC, we applied spatially stratified random sampling to select 200 sites per pollutant. Site counts per network and pollutant can be found at Appendix A. Details regarding model checkpoints, hyperparameters, and training configurations can be found in Appendix B. Sites with degenerate Mean Absolute

Scaled Error (MASE) or Continuous Ranked Probability Score (CRPS) were excluded from evaluation; details and excluded site counts are reported in Appendix C.

## 4.1. Results

*Table 1.* Pollutant-balanced overall leaderboard

| model | MASE (norm.) | CRPS (norm.) |
|---|---|---|
| **TSFMs** | | |
| Chronos-2 | 0.7929 | 0.4654 |
| Chronos-Bolt | 0.7976 | 0.4999 |
| Kairos | 1.0121 | 0.6426 |
| Moirai-1 | 0.8103 | 0.4670 |
| Moirai-2 | 0.7916 | 0.4421 |
| Sundial | 0.7977 | 0.5728 |
| TiRex | 0.7825 | 0.4553 |
| TimesFM-1.0 | 0.8298 | 0.5133 |
| TimesFM-2.0 | 0.8013 | 0.4807 |
| TimesFM-2.5 | *0.7831* | **0.4385** |
| VisionTS++ | **0.7785** | *0.4537* |
| **ML Baselines** | | |
| DLinear | 0.8999 | 0.5652 |
| DeepAR | 0.9091 | 0.4911 |
| LightGBM | 0.9268 | 0.5765 |
| PatchTST | 0.8300 | 0.4625 |
| **Statistical Baselines** | | |
| AutoETS | 0.8949 | 0.7878 |
| Seasonal Naive | 1.0000 | 1.0000 |

Table 1 reports the overall leaderboard. Following Aksu et al. (2024), scores are aggregated by taking the mean over rolling windows, then over series, then over pollutants, and finally the geometric mean over networks to account for heterogeneous difficulty levels across datasets. Results are normalized by Seasonal Naive. Detailed results individually for each network and pollutant before normalization can be found in Appendix D.

VisionTS++ (Shen et al., 2025) achieves the lowest overall MASE, followed by TiRex (Auer et al., 2025) and TimesFM-2.5. VisionTS++ also holds the top spot on each individual country. Notably, VisionTS++ operates by rendering time series as images and leveraging a vision foundation model. This cross-modal approach suggests that multimodal pretraining may confer substantial advantages over purely temporal architectures, and that visual representations of temporal patterns remain a promising direction for environmental forecasting. Among supervised baselines, PatchTST is competitive with lower ranked TSFMs, while DLinear underperforms AutoETS in aggregate, driven by severe degradation on CPCB and $SO_2$ series. Kairos (Feng et al., 2026) is the weakest TSFM overall, the only model to fall below Seasonal Naive. We believe this may be due to its architecture being optimized for much longer context windows than the 168-hour input used here.

Cross-network results (Appendix Table 48) reveal a consistent picture. TSFM relative rankings are stable across all seven networks, with VisionTS++, TiRex, and TimesFM-2.5 occupying the top positions on every leaderboard, while absolute forecasting difficulty varies substantially. AURN and EEA-France are easiest, while CPCB is hardest among well-sampled networks. On CPCB, both DLinear and PatchTST fail to improve on Seasonal Naive despite being trained on that data. This points to genuine distributional complexity rather than mere domain shift, and confirms the task is hard rather than simply unfamiliar to zero-shot models. Similarly cross-pollutant results follow a consistent hierarchy across networks. Particulate matter is most predictable, followed by $NO_2$ and CO, while $SO_2$ and $O_3$ pose the hardest challenges for all model classes. Notably, DLinear and Light-GBM exceed a normalized MASE of 1.0 on $SO_2$. Within the top tier, VisionTS++ leads on the easier pollutants (CO, $NO_2$, $PM_{10}$, $PM_{2.5}$) while TiRex leads on $O_3$ and $SO_2$.

The networks where zero-shot performance is weakest, CPCB and SINAICA, are also the ones where forecasting failures carry the greatest consequence. Indian stations in AQA-Bench record mean $PM_{2.5}$ concentrations of 57 $\mu g/m^3$ and $PM_{10}$ of 122 $\mu g/m^3$ (Table 47), roughly twelve and eight times the WHO annual guidelines respectively (World Health Organization, 2021). A model benchmarked only on European or US data would appear to work well while remaining unvalidated precisely where pollution is worst.

## 5. Conclusion and Future Work

AQA-Bench establishes TSFMs as a viable and broadly applicable approach to air quality forecasting, outperforming statistical baselines across geographies and pollutants in a zero-shot setting. Our work answers a practically important question, showing that a pretrained foundation model can outperform models optimised directly on the target series. The performance differences observed across urban profiles highlight why benchmark diversity matters. Evaluating only on cleaner networks obscures how models perform precisely where pollution is highest and the consequences of forecasting failures are greatest. AQA-Bench addresses these gaps and provides a more representative evaluation framework for diverse air quality conditions.

The gaps AQA-Bench surfaces motivate important directions for future work. Expanding geographic coverage, few-shot adaptation to high-pollution networks, evaluating multiple horizons, and modelling additional covariates for reactive pollutants are all directions we hope to pursue. In particular, exploration of cross-modal architectures following the strong performance of VisionTS++ seems to be a very promising direction to pursue. We encourage the community to do the same using our open-sourced framework.

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

## Acknowledgements

We would like to acknowledge all government entities who made this data public, as well as quotsoft.net for collecting the historical CNEMC data. We would also like to acknowledge the Anuradha and Prashanth Palakurthi Centre for Artificial Intelligence Research (APPCAIR), BITS Pilani for funding support.

## A. Monitoring Sites per Pollutant and Network

*Table 2.* Constituent datasets in AQA-Data.

| Network | Country | Source |
|---------|---------|--------|
| EPA AQS | United States | U.S. Environmental Protection Agency |
| CPCB | India | Central Pollution Control Board |
| AURN | United Kingdom | Department for Environment, Food & Rural Affairs |
| CNEMC | China | quotsoft.net |
| EEA | France, Germany | European Environment Agency |
| SINAICA | Mexico | Gobierno de México |

All data collected was present at an hourly temporal resolution, except CPCB data. CPCB data was collected at a 15 minute resolution and resampled to hourly via median aggregation. After quality filtering (requiring at least 70% valid observations and no gaps exceeding two weeks), the number of retained stations varies substantially across networks and pollutants. CNEMC has by far the largest coverage; for evaluation it was subsampled to 200 sites per pollutant via spatially stratified random sampling. CO coverage is sparse for AURN and EEA-FR, therefore results should be interpreted with caution.

*Table 3.* Number of monitoring sites per pollutant and network in AQA-Data.

| Network | CO | NO$_2$ | O$_3$ | PM$_{10}$ | PM$_{2.5}$ | SO$_2$ |
|---------|-----|------|-----|-----|------|-----|
| EPA | 90 | 212 | 488 | 220 | 445 | 212 |
| AURN | 1 | 70 | 39 | 80 | 61 | 8 |
| CPCB | 179 | 179 | 182 | 191 | 190 | 183 |
| CNEMC | 1487 | 1481 | 1488 | 1481 | 1471 | 1485 |
| EEA-FR | 5 | 219 | 211 | 204 | 131 | 37 |
| EEA-DE | 65 | 370 | 262 | 316 | 246 | 75 |
| SINAICA | 12 | 11 | 15 | 16 | 6 | 15 |
| **Total: 14,139** | | | | | | |

## B. Model Information

All experiments were conducted on a single NVIDIA RTX A5000 GPU. We used the following checkpoint variants: Moirai (Base) (Woo et al., 2024), Moirai-2 (Small) (Liu et al., 2025a), Chronos-Bolt (Base) and Chronos-2 (Base) (Ansari et al., 2024; 2025), Kairos (50M) (Feng et al., 2026), Sundial (Base, 128M) (Liu et al., 2025b), VisionTS++ (Base) (Shen et al., 2025), TimesFM-1.0 (Base, 200M), 2.0 (Base, 500M), and 2.5 (Base, 200M) (Das et al., 2024), and TiRex (Base) (Auer et al., 2025). Seasonal Naive and AutoETS were used from

the statsforecast library (Garza et al., 2022). DeepAR, Light-GBM, DLinear, and PatchTST were implemented from the AutoGluon library (Erickson et al., 2020; Shchur et al., 2023).

Data is split chronologically, the first year of data serves as the training split and the remaining two years as the evaluation split. As the majority of models require no training, a larger test split better captures zero-shot performance across seasonal variation. All evaluated TSFMs support native probabilistic forecasting and were used without any weight updates or adaptation to AQA-Data. ML baselines were trained per dataset per pollutant on the training split and evaluated on the test split. Statistical baselines are fitted directly on each test window, as is standard for this model class.

AutoETS was set to AZA (additive errors, automatic trend, additive seasonality) to prevent multiplicative zero errors arising from near-zero pollutant values. DeepAR, PatchTST and DLinear were trained for 100 epochs, with an early stopping patience of 10. LightGBM was trained for 1000 boost rounds, with the same early stopping patience.

## C. Excluded Sites

MASE is computed using a daily seasonal naive baseline (lag 24) over each 168-hour context window. Sites with a mean MASE or CRPS over all models exceeding 50 were excluded as degenerate, typically arising from near-constant or near-zero series where the Seasonal Naive denominator collapses. Exclusion was applied independently per pollutant and primarily affected $SO_2$ in the EPA and EEA-DE datasets. Exact excluded site counts are reported below. No exclusion was applied for MAE or RMSE.

*Table 4.* Sites excluded per network and pollutant due to degenerate MASE or CRPS scores (mean > 50 across all models).

| Network | CO | NO$_2$ | SO$_2$ | Total |
|---|---|---|---|---|
| CPCB | 1 | – | – | 1 |
| EEA-DE | – | 2 | 20 | 22 |
| EPA | 2 | – | 53 | 55 |
| **All others** | – | – | – | 0 |

## D. Per-Pollutant Results

Results are organised by monitoring network (alphabetical), with all six pollutants shown per network.

### D.1. AURN

*Table 5.* CO leaderboard — AURN

| model | MASE | CRPS | MAE | RMSE |
|---|---|---|---|---|
| **TSFMs** | | | | |
| Chronos-2 | 0.8234 | 0.2082 | 0.0491 | 0.0886 |
| Chronos-Bolt | 0.8380 | 0.2113 | 0.0500 | 0.0914 |
| Kairos | 0.9203 | 0.2376 | 0.0558 | 0.0982 |
| Moirai-1 | 0.8451 | 0.2155 | 0.0502 | 0.0910 |
| Moirai-2 | 0.7988 | 0.2011 | 0.0473 | *0.0854* |
| Sundial | 0.8210 | 0.2241 | 0.0492 | 0.0881 |
| TiRex | 0.7932 | 0.2003 | 0.0467 | 0.0850 |
| TimesFM-1.0 | 0.8306 | 0.2090 | 0.0488 | 0.0858 |
| TimesFM-2.0 | 0.8168 | 0.2068 | 0.0482 | 0.0870 |
| TimesFM-2.5 | 0.7980 | *0.2003* | 0.0473 | 0.0857 |
| VisionTS++ | **0.7881** | 0.1983 | **0.0465** | **0.0849** |
| **ML Baselines** | | | | |
| DLinear | 0.8928 | 0.2312 | 0.0545 | 0.0953 |
| DeepAR | 0.8960 | 0.2297 | 0.0546 | 0.1005 |
| LightGBM | 1.0031 | 0.2411 | 0.0588 | 0.1048 |
| PatchTST | *0.7951* | **0.1939** | *0.0472* | 0.0913 |
| **Statistical Baselines** | | | | |
| AutoETS | 0.9436 | 0.2508 | 0.0571 | 0.0967 |
| Seasonal Naive | 1.0884 | 0.3096 | 0.0667 | 0.1222 |

*Table 6.* NO2 leaderboard — AURN

| model | MASE | CRPS | MAE | RMSE |
|---|---|---|---|---|
| **TSFMs** | | | | |
| Chronos-2 | 0.8363 | 0.3816 | 7.1325 | 10.5645 |
| Chronos-Bolt | 0.8256 | 0.3810 | *7.0254* | 10.3525 |
| Kairos | 1.0233 | 0.4626 | 8.9212 | 12.4706 |
| Moirai-1 | 0.8514 | 0.3881 | 7.2730 | 10.6402 |
| Moirai-2 | 0.8323 | 0.3741 | 7.0968 | 10.4744 |
| Sundial | 0.8341 | 0.4260 | 7.1247 | 10.3836 |
| TiRex | 0.8319 | 0.3798 | 7.1207 | 10.5456 |
| TimesFM-1.0 | 0.8432 | 0.3850 | 7.1944 | 10.6455 |
| TimesFM-2.0 | 0.8297 | 0.3724 | 7.0633 | 10.5288 |
| TimesFM-2.5 | *0.8242* | *0.3688* | 7.0258 | 10.4115 |
| VisionTS++ | **0.8143** | 0.3684 | 6.9311 | *10.2834* |
| **ML Baselines** | | | | |
| DLinear | 0.8282 | 0.3814 | 7.0337 | 10.2002 |
| DeepAR | 0.8524 | **0.3643** | 7.1747 | 10.8434 |
| LightGBM | 0.8507 | 0.3896 | 7.2321 | 10.6544 |
| PatchTST | 0.8156 | 0.3697 | **6.9258** | **10.1628** |
| **Statistical Baselines** | | | | |
| AutoETS | 0.9368 | 0.4591 | 7.9941 | 11.1956 |
| Seasonal Naive | 1.0634 | 0.5459 | 9.1065 | 13.1011 |

*Table 7.* Ozone leaderboard — AURN

| model | MASE | CRPS | MAE | RMSE |
|---|---|---|---|---|
| **TSFMs** | | | | |
| Chronos-2 | 0.8091 | 0.2173 | 11.8232 | 15.9042 |
| Chronos-Bolt | 0.8135 | 0.2182 | 11.8944 | 15.9953 |
| Kairos | 1.0052 | 0.2715 | 14.6526 | 19.3464 |
| Moirai-1 | 0.8310 | 0.2235 | 12.1426 | 16.1774 |
| Moirai-2 | 0.8133 | 0.2198 | 11.9039 | 15.9355 |
| Sundial | 0.7964 | 0.2378 | 11.6216 | *15.4175* |
| TiRex | **0.7799** | **0.2116** | **11.3724** | **15.3199** |
| TimesFM-1.0 | 0.8578 | 0.2421 | 12.5846 | 16.7826 |
| TimesFM-2.0 | 0.7953 | 0.2201 | 11.6071 | 15.5203 |
| TimesFM-2.5 | 0.7866 | *0.2152* | 11.4601 | 15.3572 |
| VisionTS++ | *0.7930* | 0.2132 | *11.5670* | 15.4976 |
| **ML Baselines** | | | | |
| DLinear | 0.8048 | 0.2328 | 11.7243 | 15.6214 |
| DeepAR | 0.8175 | 0.2633 | 11.9502 | 15.8906 |
| LightGBM | 0.8591 | 0.2600 | 12.5791 | 16.3401 |
| PatchTST | 0.8153 | 0.2414 | 11.8864 | 15.8026 |
| **Statistical Baselines** | | | | |
| AutoETS | 0.8976 | 0.2511 | 13.1541 | 17.1105 |
| Seasonal Naive | 1.0503 | 0.2999 | 15.4741 | 20.3091 |

*Table 9.* PM2.5 leaderboard — AURN

| model | MASE | CRPS | MAE | RMSE |
|---|---|---|---|---|
| **TSFMs** | | | | |
| Chronos-2 | 0.8385 | 0.3696 | 3.1806 | 5.1566 |
| Chronos-Bolt | 0.8523 | 0.3772 | 3.2286 | 5.2664 |
| Kairos | 0.9130 | 0.4201 | 3.5345 | 5.5554 |
| Moirai-1 | 0.8438 | 0.3789 | 3.2118 | 5.2348 |
| Moirai-2 | 0.8253 | 0.3582 | 3.1172 | 5.0622 |
| Sundial | 0.8328 | 0.4042 | 3.1815 | 5.0857 |
| TiRex | *0.8196* | 0.3618 | *3.1056* | 5.0375 |
| TimesFM-1.0 | 0.8627 | 0.3914 | 3.2936 | 5.2673 |
| TimesFM-2.0 | 0.8302 | 0.3710 | 3.1588 | 5.0958 |
| TimesFM-2.5 | **0.8172** | **0.3579** | **3.0898** | *5.0191* |
| VisionTS++ | 0.8180 | *0.3615* | 3.0913 | 5.0151 |
| **ML Baselines** | | | | |
| DLinear | 0.8542 | 0.4022 | 3.3123 | 5.1084 |
| DeepAR | 0.8490 | 0.3845 | 3.2414 | 5.2341 |
| LightGBM | 0.9087 | 0.3829 | 3.4312 | 5.6275 |
| PatchTST | 0.8258 | 0.3630 | 3.1088 | **4.9867** |
| **Statistical Baselines** | | | | |
| AutoETS | 0.9254 | 0.4397 | 3.5609 | 5.5985 |
| Seasonal Naive | 1.1003 | 0.5557 | 4.2612 | 6.7161 |

*Table 8.* PM10 leaderboard — AURN

| model | MASE | CRPS | MAE | RMSE |
|---|---|---|---|---|
| **TSFMs** | | | | |
| Chronos-2 | 0.8297 | 0.3434 | 5.2985 | 8.3368 |
| Chronos-Bolt | 0.8361 | 0.3483 | 5.3399 | 8.4259 |
| Kairos | 0.8857 | 0.3737 | 5.7243 | 8.8172 |
| Moirai-1 | 0.8337 | 0.3488 | 5.3336 | 8.4093 |
| Moirai-2 | 0.8269 | 0.3346 | 5.2644 | 8.3249 |
| Sundial | 0.8229 | 0.3732 | 5.2762 | 8.2389 |
| TiRex | 0.8182 | 0.3396 | 5.2249 | 8.3025 |
| TimesFM-1.0 | 0.8462 | 0.3551 | 5.4182 | 8.4812 |
| TimesFM-2.0 | 0.8315 | 0.3464 | 5.3194 | 8.3669 |
| TimesFM-2.5 | *0.8149* | **0.3326** | *5.1859* | 8.2257 |
| VisionTS++ | 0.8107 | *0.3357* | 5.1600 | *8.1505* |
| **ML Baselines** | | | | |
| DLinear | 0.8176 | 0.3497 | 5.2826 | 8.1483 |
| DeepAR | 0.8272 | 0.3403 | 5.2679 | 8.3788 |
| LightGBM | 0.8723 | 0.3507 | 5.5308 | 8.7145 |
| PatchTST | **0.8086** | 0.3381 | **5.1435** | **8.0364** |
| **Statistical Baselines** | | | | |
| AutoETS | 0.9066 | 0.4005 | 5.8721 | 8.9064 |
| Seasonal Naive | 1.0750 | 0.4934 | 6.9671 | 10.6799 |

*Table 10.* SO2 leaderboard — AURN

| model | MASE | CRPS | MAE | RMSE |
|---|---|---|---|---|
| **TSFMs** | | | | |
| Chronos-2 | 1.0549 | 0.5089 | 0.7314 | 3.7773 |
| Chronos-Bolt | 1.0399 | 0.4979 | 0.7272 | 3.7711 |
| Kairos | 1.0768 | 0.5594 | 0.7608 | 3.7498 |
| Moirai-1 | 1.0407 | 0.4764 | 0.7090 | 3.7182 |
| Moirai-2 | 1.0371 | 0.4640 | 0.7003 | 3.7008 |
| Sundial | 1.0253 | 0.5818 | 0.7445 | **3.6520** |
| TiRex | *1.0201* | 0.4723 | 0.6963 | 3.7210 |
| TimesFM-1.0 | 1.0323 | 0.4692 | 0.6985 | *3.6541* |
| TimesFM-2.0 | 1.0461 | 0.4733 | *0.6948* | 3.6538 |
| TimesFM-2.5 | 1.0289 | *0.4592* | 0.6977 | 3.6845 |
| VisionTS++ | 1.0268 | 0.4796 | 0.7056 | 3.7106 |
| **ML Baselines** | | | | |
| DLinear | 1.0607 | 0.5956 | 0.8311 | 3.6892 |
| DeepAR | **1.0071** | **0.4358** | **0.6777** | 3.6793 |
| LightGBM | 1.0310 | 0.4958 | 0.7098 | 3.6604 |
| PatchTST | 1.0187 | 0.4438 | 0.6836 | 3.6699 |
| **Statistical Baselines** | | | | |
| AutoETS | 1.2048 | 1.0229 | 1.0786 | 4.2029 |
| Seasonal Naive | 1.3008 | 1.2936 | 1.0486 | 4.9559 |

**D.2. CNEMC**

*Table 11.* CO leaderboard — CNEMC

| model | MASE | CRPS | MAE | RMSE |
|---|---|---|---|---|
| **TSFMs** | | | | |
| Chronos-2 | 0.8536 | 0.1781 | 0.1379 | 0.2191 |
| Chronos-Bolt | 0.8583 | 0.1796 | 0.1386 | 0.2202 |
| Kairos | 1.0337 | 0.2194 | 0.1680 | 0.2504 |
| Moirai-1 | 0.8642 | 0.1821 | 0.1391 | 0.2192 |
| Moirai-2 | 0.8458 | *0.1759* | 0.1360 | 0.2139 |
| Sundial | 0.8585 | 0.1936 | 0.1386 | 0.2138 |
| TiRex | *0.8432* | 0.1771 | 0.1364 | 0.2162 |
| TimesFM-1.0 | 0.8905 | 0.1879 | 0.1433 | 0.2217 |
| TimesFM-2.0 | 0.8664 | 0.1812 | 0.1392 | 0.2181 |
| TimesFM-2.5 | 0.8416 | **0.1751** | 0.1351 | *0.2129* |
| VisionTS++ | **0.8351** | 0.1752 | **0.1344** | 0.2118 |
| **ML Baselines** | | | | |
| DLinear | 0.8698 | 0.1865 | 0.1398 | 0.2131 |
| DeepAR | 0.9161 | 0.1879 | 0.1473 | 0.2350 |
| LightGBM | 0.9122 | 0.1893 | 0.1455 | 0.2238 |
| PatchTST | 0.8472 | 0.1762 | *0.1356* | **0.2109** |
| **Statistical Baselines** | | | | |
| AutoETS | 0.9559 | 0.2100 | 0.1563 | 0.2331 |
| Seasonal Naive | 1.0727 | 0.2553 | 0.1795 | 0.2805 |

*Table 13.* Ozone leaderboard — CNEMC

| model | MASE | CRPS | MAE | RMSE |
|---|---|---|---|---|
| **TSFMs** | | | | |
| Chronos-2 | 0.8515 | 0.2380 | 17.5516 | 24.4027 |
| Chronos-Bolt | 0.8465 | *0.2372* | 17.4513 | 24.2806 |
| Kairos | 1.4097 | 0.3832 | 28.7210 | 38.4039 |
| Moirai-1 | 0.8823 | 0.2524 | 18.2204 | 25.0719 |
| Moirai-2 | 0.8648 | 0.2427 | 17.8240 | 24.6095 |
| Sundial | 0.8527 | 0.2600 | 17.5337 | *23.8977* |
| TiRex | 0.8412 | 0.2362 | *17.3516* | 24.1559 |
| TimesFM-1.0 | 0.9110 | 0.2586 | 18.7911 | 25.8213 |
| TimesFM-2.0 | 0.8620 | 0.2454 | 17.7385 | 24.3700 |
| TimesFM-2.5 | *0.8428* | 0.2377 | 17.3373 | 23.8830 |
| VisionTS++ | 0.8445 | **0.2361** | 17.4011 | 24.0907 |
| **ML Baselines** | | | | |
| DLinear | 0.8904 | 0.2466 | 18.1854 | 24.5150 |
| DeepAR | 0.8872 | 0.2667 | 18.2978 | 24.6612 |
| LightGBM | 0.8760 | 0.2448 | 17.9704 | 24.1388 |
| PatchTST | **0.8303** | 0.2388 | **17.0237** | **23.0460** |
| **Statistical Baselines** | | | | |
| AutoETS | 0.9649 | 0.2798 | 19.9135 | 26.3167 |
| Seasonal Naive | 1.0495 | 0.3095 | 21.8108 | 29.5152 |

*Table 12.* NO2 leaderboard — CNEMC

| model | MASE | CRPS | MAE | RMSE |
|---|---|---|---|---|
| **TSFMs** | | | | |
| Chronos-2 | 0.8188 | 0.2914 | 7.4805 | 11.3595 |
| Chronos-Bolt | 0.8208 | 0.2952 | 7.5027 | 11.4165 |
| Kairos | 1.1380 | 0.4147 | 10.4930 | 14.7121 |
| Moirai-1 | 0.8287 | 0.2990 | 7.5771 | 11.3383 |
| Moirai-2 | *0.8082* | *0.2871* | *7.3716* | 11.1477 |
| Sundial | 0.8332 | 0.3269 | 7.6515 | 11.3152 |
| TiRex | 0.8204 | 0.2931 | 7.5177 | 11.3714 |
| TimesFM-1.0 | 0.8432 | 0.3069 | 7.7486 | 11.5995 |
| TimesFM-2.0 | 0.8226 | 0.2936 | 7.5105 | 11.3686 |
| TimesFM-2.5 | 0.8098 | 0.2873 | 7.3840 | 11.1352 |
| VisionTS++ | **0.8003** | 0.2856 | **7.3098** | *11.0871* |
| **ML Baselines** | | | | |
| DLinear | 0.8184 | 0.2978 | 7.4954 | 11.0636 |
| DeepAR | 0.8315 | 0.2942 | 7.5725 | 11.3417 |
| LightGBM | 0.8548 | 0.3145 | 7.8826 | 11.5634 |
| PatchTST | 0.8035 | **0.2837** | 7.3271 | **10.9889** |
| **Statistical Baselines** | | | | |
| AutoETS | 0.9359 | 0.3595 | 8.6695 | 12.3952 |
| Seasonal Naive | 1.0563 | 0.4285 | 9.8504 | 14.7252 |

*Table 14.* PM10 leaderboard — CNEMC

| model | MASE | CRPS | MAE | RMSE |
|---|---|---|---|---|
| **TSFMs** | | | | |
| Chronos-2 | 0.8576 | 0.2957 | 21.9493 | 51.5332 |
| Chronos-Bolt | 0.8746 | 0.3041 | 22.7016 | 55.8858 |
| Kairos | 0.9850 | 0.3582 | 25.1144 | 52.3456 |
| Moirai-1 | 0.8616 | 0.3031 | 22.0436 | 49.5990 |
| Moirai-2 | 0.8449 | 0.2886 | 21.3591 | *48.5309* |
| Sundial | 0.8623 | 0.3296 | 22.1074 | 48.7871 |
| TiRex | 0.8397 | 0.2921 | *21.2632* | 48.2206 |
| TimesFM-1.0 | 0.8755 | 0.3102 | 22.3289 | 49.3871 |
| TimesFM-2.0 | 0.8575 | 0.3018 | 21.8109 | 48.6662 |
| TimesFM-2.5 | *0.8380* | *0.2887* | 21.1425 | **48.0782** |
| VisionTS++ | 0.8353 | 0.2897 | 21.2954 | 48.6429 |
| **ML Baselines** | | | | |
| DLinear | 0.8580 | 0.3019 | 22.3329 | 49.1409 |
| DeepAR | 0.8828 | 0.3090 | 22.6221 | 50.0531 |
| LightGBM | 0.9096 | 0.3166 | 23.4496 | 51.0299 |
| PatchTST | **0.8335** | **0.2809** | **21.1055** | 48.6913 |
| **Statistical Baselines** | | | | |
| AutoETS | 0.9604 | 0.3771 | 25.9448 | 54.8411 |
| Seasonal Naive | 1.1213 | 0.4713 | 30.2852 | 65.9397 |

*Table 15.* PM2.5 leaderboard — CNEMC

| model | MASE | CRPS | MAE | RMSE |
|---|---|---|---|---|
| **TSFMs** | | | | |
| Chronos-2 | 0.8343 | 0.3333 | 11.5998 | 23.2657 |
| Chronos-Bolt | 0.8493 | 0.3429 | 11.8942 | 24.5609 |
| Kairos | 0.9669 | 0.4160 | 13.5590 | 23.5243 |
| Moirai-1 | 0.8385 | 0.3449 | 11.6368 | 21.7274 |
| Moirai-2 | 0.8240 | **0.3278** | 11.3236 | 21.0828 |
| Sundial | 0.8402 | 0.3766 | 11.6636 | 21.2644 |
| TiRex | *0.8190* | 0.3316 | *11.3072* | 21.1466 |
| TimesFM-1.0 | 0.8561 | 0.3567 | 11.8994 | 21.7923 |
| TimesFM-2.0 | 0.8341 | 0.3424 | 11.5130 | 21.2348 |
| TimesFM-2.5 | 0.8151 | 0.3284 | **11.1904** | **20.8240** |
| VisionTS++ | **0.8137** | *0.3295* | 11.2294 | *21.1204* |
| **ML Baselines** | | | | |
| DLinear | 0.8438 | 0.3560 | 11.8049 | 21.2023 |
| DeepAR | 0.8467 | 0.3402 | 11.6795 | 21.7003 |
| LightGBM | 0.9128 | 0.3613 | 12.6479 | 23.0049 |
| PatchTST | 0.8337 | 0.3356 | 11.5250 | 21.3781 |
| **Statistical Baselines** | | | | |
| AutoETS | 0.9317 | 0.4187 | 13.3167 | 24.0191 |
| Seasonal Naive | 1.0977 | 0.5284 | 15.8193 | 28.9843 |

**D.3. CPCB**

*Table 17.* CO leaderboard — CPCB

| model | MASE | CRPS | MAE | RMSE |
|---|---|---|---|---|
| **TSFMs** | | | | |
| Chronos-2 | 0.9244 | 0.2618 | 0.2482 | 0.4584 |
| Chronos-Bolt | 0.9419 | 0.2700 | 0.2515 | 0.4678 |
| Kairos | 1.2723 | 0.3652 | 0.3325 | 0.5519 |
| Moirai-1 | 0.9621 | 0.2810 | 0.2547 | 0.4565 |
| Moirai-2 | 0.9172 | **0.2555** | 0.2452 | 0.4431 |
| Sundial | 0.9103 | 0.2898 | *0.2428* | **0.4342** |
| TiRex | 0.9060 | *0.2586* | 0.2434 | 0.4453 |
| TimesFM-1.0 | 0.9476 | 0.2844 | 0.2512 | 0.4502 |
| TimesFM-2.0 | 0.9228 | 0.2716 | 0.2454 | 0.4454 |
| TimesFM-2.5 | **0.9043** | 0.2612 | **0.2414** | 0.4380 |
| VisionTS++ | *0.9074* | 0.2572 | 0.2421 | 0.4419 |
| **ML Baselines** | | | | |
| DLinear | 1.0502 | 0.3095 | 0.2581 | 0.4456 |
| DeepAR | 0.9918 | 0.2587 | 0.2507 | 0.4475 |
| LightGBM | 0.9945 | 0.3036 | 0.2534 | 0.4428 |
| PatchTST | 0.9779 | 0.2683 | 0.2476 | *0.4413* |
| **Statistical Baselines** | | | | |
| AutoETS | 1.0189 | 0.3330 | 0.2751 | 0.4686 |
| Seasonal Naive | 1.1246 | 0.4027 | 0.3069 | 0.5545 |

*Table 16.* SO2 leaderboard — CNEMC

| model | MASE | CRPS | MAE | RMSE |
|---|---|---|---|---|
| **TSFMs** | | | | |
| Chronos-2 | 0.9330 | 0.2042 | 2.1744 | 4.5784 |
| Chronos-Bolt | 0.9395 | 0.2059 | 2.1863 | 4.6160 |
| Kairos | 1.0599 | 0.2352 | 2.5095 | 4.9840 |
| Moirai-1 | 0.9437 | 0.2064 | 2.1916 | 4.6164 |
| Moirai-2 | 0.9295 | 0.2013 | 2.1559 | 4.5579 |
| Sundial | 0.9452 | 0.2263 | 2.2157 | 4.5111 |
| TiRex | 0.9230 | 0.2007 | *2.1433* | 4.5606 |
| TimesFM-1.0 | 0.9740 | 0.2113 | 2.2277 | 4.6152 |
| TimesFM-2.0 | 0.9470 | 0.2037 | 2.1788 | 4.6132 |
| TimesFM-2.5 | *0.9262* | **0.1991** | 2.1410 | 4.5618 |
| VisionTS++ | **0.9204** | *0.2009* | **2.1295** | **4.5077** |
| **ML Baselines** | | | | |
| DLinear | 1.1207 | 0.2282 | 2.3040 | *4.5377* |
| DeepAR | 1.1635 | 0.2082 | 2.2860 | 4.8613 |
| LightGBM | 1.1722 | 0.2161 | 2.2416 | 4.5945 |
| PatchTST | 1.0139 | 0.2033 | 2.1704 | 4.5816 |
| **Statistical Baselines** | | | | |
| AutoETS | 1.0628 | 0.2658 | 2.5879 | 4.8624 |
| Seasonal Naive | 1.1584 | 0.3282 | 2.8828 | 5.9823 |

*Table 18.* NO2 leaderboard — CPCB

| model | MASE | CRPS | MAE | RMSE |
|---|---|---|---|---|
| **TSFMs** | | | | |
| Chronos-2 | 1.0906 | 0.1619 | 4.9475 | 10.2941 |
| Chronos-Bolt | 1.0990 | 0.1685 | 4.9901 | 10.4766 |
| Kairos | 1.4645 | 0.2509 | 7.4305 | 13.2823 |
| Moirai-1 | 1.1209 | 0.1756 | 5.1044 | 10.3942 |
| Moirai-2 | 1.0900 | *0.1616* | *4.9067* | 10.0536 |
| Sundial | 1.1141 | 0.1922 | 5.0875 | 10.1572 |
| TiRex | 1.0854 | 0.1638 | 4.9244 | 10.1495 |
| TimesFM-1.0 | 1.1952 | 0.1763 | 5.1509 | 10.4106 |
| TimesFM-2.0 | 1.1318 | 0.1699 | 5.0297 | 10.2901 |
| TimesFM-2.5 | *1.0865* | **0.1593** | 4.8959 | **10.0528** |
| VisionTS++ | **1.0806** | 0.1609 | **4.8543** | *10.0593* |
| **ML Baselines** | | | | |
| DLinear | 1.7975 | 0.2402 | 5.2552 | 10.2494 |
| DeepAR | 2.7737 | 0.2075 | 5.5075 | 10.6324 |
| LightGBM | 1.6227 | 0.2091 | 5.2668 | 10.5148 |
| PatchTST | 1.5473 | 0.1839 | 5.2061 | 10.5882 |
| **Statistical Baselines** | | | | |
| AutoETS | 1.2158 | 0.3116 | 5.7697 | 10.9152 |
| Seasonal Naive | 1.3420 | 0.3745 | 6.3035 | 12.8719 |

*Table 19.* Ozone leaderboard — CPCB

| model | MASE | CRPS | MAE | RMSE |
|---|---|---|---|---|
| **TSFMs** | | | | |
| Chronos-2 | 0.9423 | *0.2266* | *7.4354* | 13.7858 |
| Chronos-Bolt | 0.9739 | 0.2337 | 7.5221 | 13.8563 |
| Kairos | 2.6636 | 0.4313 | 14.7043 | 23.7030 |
| Moirai-1 | 1.0567 | 0.2422 | 7.8403 | 14.2222 |
| Moirai-2 | 0.9789 | 0.2272 | 7.5333 | 13.8471 |
| Sundial | 0.9909 | 0.2591 | 7.6890 | 13.7808 |
| TiRex | **0.9391** | 0.2263 | 7.3793 | 13.6980 |
| TimesFM-1.0 | 1.1591 | 0.2536 | 8.1274 | 14.5997 |
| TimesFM-2.0 | 1.0246 | 0.2426 | 7.7659 | 14.1007 |
| TimesFM-2.5 | 0.9578 | 0.2279 | 7.4962 | *13.7385* |
| VisionTS++ | *0.9534* | **0.2240** | **7.3577** | **13.6291** |
| **ML Baselines** | | | | |
| DLinear | 1.5990 | 0.2816 | 7.9252 | 13.8600 |
| DeepAR | 2.1098 | 0.2424 | 7.9193 | 14.1435 |
| LightGBM | 1.0854 | 0.2533 | 7.7494 | 14.0969 |
| PatchTST | 1.2962 | 0.2388 | 7.7170 | 13.8499 |
| **Statistical Baselines** | | | | |
| AutoETS | 1.0750 | 0.3011 | 8.6509 | 14.6753 |
| Seasonal Naive | 1.1367 | 0.3663 | 9.1527 | 16.8007 |

*Table 21.* PM2.5 leaderboard — CPCB

| model | MASE | CRPS | MAE | RMSE |
|---|---|---|---|---|
| **TSFMs** | | | | |
| Chronos-2 | 0.8530 | 0.2604 | 16.5335 | 32.1943 |
| Chronos-Bolt | 0.8643 | 0.3098 | 16.7812 | 34.4363 |
| Kairos | 1.1025 | 0.4086 | 22.2215 | 37.7633 |
| Moirai-1 | 0.8680 | 0.2845 | 16.7435 | 31.3103 |
| Moirai-2 | 0.8460 | *0.2560* | *16.2858* | *30.6680* |
| Sundial | 0.8630 | 0.3287 | 16.6975 | 30.7959 |
| TiRex | *0.8444* | 0.2658 | 16.3108 | 30.9252 |
| TimesFM-1.0 | 0.8780 | 0.3090 | 16.9908 | 31.6615 |
| TimesFM-2.0 | 0.8582 | 0.2815 | 16.4874 | 30.9618 |
| TimesFM-2.5 | 0.8429 | **0.2550** | 16.2342 | 30.5685 |
| VisionTS++ | **0.8324** | 0.2560 | **16.0400** | **30.4889** |
| **ML Baselines** | | | | |
| DLinear | 0.9364 | 0.3477 | 16.8454 | 30.7986 |
| DeepAR | 0.9807 | 0.3353 | 16.9515 | 32.2515 |
| LightGBM | 0.9732 | 0.5845 | 17.8979 | 32.1285 |
| PatchTST | 0.9696 | 0.3268 | 16.8673 | 31.5649 |
| **Statistical Baselines** | | | | |
| AutoETS | 0.9533 | 0.4163 | 18.9739 | 33.8174 |
| Seasonal Naive | 1.0807 | 0.5142 | 21.3757 | 40.2379 |

*Table 20.* PM10 leaderboard — CPCB

| model | MASE | CRPS | MAE | RMSE |
|---|---|---|---|---|
| **TSFMs** | | | | |
| Chronos-2 | 0.8702 | 0.2302 | 32.4988 | 54.4752 |
| Chronos-Bolt | 0.8785 | 0.2344 | 32.7621 | 55.6797 |
| Kairos | 1.1371 | 0.3002 | 42.8265 | 65.5376 |
| Moirai-1 | 0.8913 | 0.2379 | 33.0893 | 54.1614 |
| Moirai-2 | 0.8636 | *0.2284* | 32.1472 | 53.0072 |
| Sundial | 0.8816 | 0.2566 | 32.9812 | 53.3992 |
| TiRex | *0.8632* | 0.2312 | 32.2666 | 53.4504 |
| TimesFM-1.0 | 0.9162 | 0.2414 | 33.4620 | 54.7526 |
| TimesFM-2.0 | 0.8756 | 0.2335 | 32.5497 | 53.7138 |
| TimesFM-2.5 | 0.8600 | 0.2276 | 32.0420 | 52.9477 |
| VisionTS++ | **0.8495** | **0.2263** | **31.6192** | **52.5272** |
| **ML Baselines** | | | | |
| DLinear | 1.3259 | 0.2420 | 32.8112 | 52.9855 |
| DeepAR | 1.2195 | 0.2323 | 33.3015 | 55.2323 |
| LightGBM | 1.2375 | 0.2570 | 34.2894 | 54.6240 |
| PatchTST | 1.5061 | 0.2286 | *32.1340* | *52.9582* |
| **Statistical Baselines** | | | | |
| AutoETS | 0.9688 | 0.2758 | 36.8008 | 57.6024 |
| Seasonal Naive | 1.0898 | 0.3289 | 41.5503 | 67.9091 |

*Table 22.* SO2 leaderboard — CPCB

| model | MASE | CRPS | MAE | RMSE |
|---|---|---|---|---|
| **TSFMs** | | | | |
| Chronos-2 | 1.1375 | 0.2115 | 3.1569 | 6.7563 |
| Chronos-Bolt | 1.1596 | 0.2149 | 3.2104 | 6.8318 |
| Kairos | 1.3247 | 0.2452 | 3.6668 | 7.1336 |
| Moirai-1 | 1.1886 | 0.2173 | 3.2560 | 6.6733 |
| Moirai-2 | 1.1344 | 0.2055 | 3.1239 | 6.4565 |
| Sundial | 1.1305 | 0.2269 | 3.1224 | **6.3206** |
| TiRex | **1.1224** | 0.2065 | 3.0898 | 6.4830 |
| TimesFM-1.0 | 1.1911 | 0.2188 | 3.2356 | 6.5849 |
| TimesFM-2.0 | 1.1516 | 0.2118 | 3.1552 | 6.5341 |
| TimesFM-2.5 | 1.1250 | **0.2041** | **3.0852** | 6.4062 |
| VisionTS++ | *1.1302* | 0.2065 | *3.0977* | *6.4233* |
| **ML Baselines** | | | | |
| DLinear | 1.4425 | 0.2472 | 3.3913 | 6.5172 |
| DeepAR | 1.5580 | 0.2273 | 3.4437 | 6.7332 |
| LightGBM | 1.2072 | 0.2211 | 3.2386 | 6.5352 |
| PatchTST | 1.2174 | 0.2104 | 3.1911 | 6.5207 |
| **Statistical Baselines** | | | | |
| AutoETS | 1.2292 | 0.2596 | 3.5339 | 6.8045 |
| Seasonal Naive | 1.3528 | 0.3217 | 3.9838 | 8.1470 |

## D.4. EEA-Germany

*Table 23.* CO leaderboard — EEA DE

| model | MASE | CRPS | MAE | RMSE |
|---|---|---|---|---|
| **TSFMs** | | | | |
| Chronos-2 | 0.8426 | 0.1798 | 0.0644 | 0.1072 |
| Chronos-Bolt | 0.8393 | 0.1798 | 0.0639 | 0.1063 |
| Kairos | 1.0330 | 0.2181 | 0.0786 | 0.1207 |
| Moirai-1 | 0.8781 | 0.1875 | 0.0668 | 0.1091 |
| Moirai-2 | 0.8530 | 0.1808 | 0.0650 | 0.1068 |
| Sundial | 0.8436 | 0.1959 | 0.0645 | *0.1049* |
| TiRex | 0.8330 | *0.1783* | *0.0637* | 0.1057 |
| TimesFM-1.0 | 0.8580 | 0.1839 | 0.0654 | 0.1073 |
| TimesFM-2.0 | 0.8437 | 0.1795 | 0.0642 | 0.1064 |
| TimesFM-2.5 | *0.8371* | 0.1771 | 0.0636 | 0.1051 |
| VisionTS++ | **0.8252** | **0.1752** | **0.0628** | **0.1042** |
| **ML Baselines** | | | | |
| DLinear | 0.8841 | 0.1935 | 0.0675 | 0.1053 |
| DeepAR | 0.9102 | 0.1878 | 0.0687 | 0.1133 |
| LightGBM | 0.9318 | 0.1920 | 0.0695 | 0.1137 |
| PatchTST | 0.8626 | 0.1832 | 0.0653 | 0.1044 |
| **Statistical Baselines** | | | | |
| AutoETS | 0.9533 | 0.2120 | 0.0736 | 0.1144 |
| Seasonal Naive | 1.0646 | 0.2475 | 0.0818 | 0.1324 |

*Table 25.* Ozone leaderboard — EEA DE

| model | MASE | CRPS | MAE | RMSE |
|---|---|---|---|---|
| **TSFMs** | | | | |
| Chronos-2 | 0.8090 | *0.2326* | 12.5185 | 16.8718 |
| Chronos-Bolt | 0.8078 | 0.2346 | 12.4934 | 16.8510 |
| Kairos | 1.2103 | 0.3442 | 18.9412 | 25.1775 |
| Moirai-1 | 0.8362 | 0.2440 | 12.9755 | 17.2611 |
| Moirai-2 | 0.8272 | 0.2440 | 12.8443 | 17.2177 |
| Sundial | 0.8108 | 0.2651 | 12.5311 | *16.5115* |
| TiRex | **0.7898** | **0.2318** | **12.2285** | 16.4471 |
| TimesFM-1.0 | 0.8606 | 0.2615 | 13.3800 | 17.7111 |
| TimesFM-2.0 | 0.8090 | 0.2412 | 12.5215 | 16.6099 |
| TimesFM-2.5 | 0.7949 | 0.2336 | 12.3024 | **16.3958** |
| VisionTS++ | *0.8005* | 0.2323 | *12.3883* | 16.5657 |
| **ML Baselines** | | | | |
| DLinear | 0.8216 | 0.2504 | 12.6760 | 16.5773 |
| DeepAR | 0.8385 | 0.2711 | 12.9442 | 16.9954 |
| LightGBM | 1.0149 | 0.3021 | 15.5326 | 19.3793 |
| PatchTST | 0.8158 | 0.2610 | 12.6040 | 16.5439 |
| **Statistical Baselines** | | | | |
| AutoETS | 0.9091 | 0.2740 | 14.0778 | 18.1145 |
| Seasonal Naive | 1.0482 | 0.3248 | 16.3203 | 21.3005 |

*Table 24.* NO2 leaderboard — EEA DE

| model | MASE | CRPS | MAE | RMSE |
|---|---|---|---|---|
| **TSFMs** | | | | |
| Chronos-2 | 0.8219 | 0.3220 | 5.3819 | 7.8577 |
| Chronos-Bolt | 0.8168 | 0.3229 | 5.3448 | 7.7813 |
| Kairos | 1.0379 | 0.4088 | 6.8216 | 9.3895 |
| Moirai-1 | 0.8443 | 0.3331 | 5.5615 | 8.0232 |
| Moirai-2 | 0.8302 | 0.3215 | 5.4656 | 7.9456 |
| Sundial | 0.8284 | 0.3592 | 5.4487 | 7.8318 |
| TiRex | 0.8227 | 0.3237 | 5.4129 | 7.8947 |
| TimesFM-1.0 | 0.8350 | 0.3331 | 5.4824 | 7.9628 |
| TimesFM-2.0 | 0.8208 | 0.3209 | 5.3786 | 7.8830 |
| TimesFM-2.5 | *0.8171* | 0.3171 | *5.3644* | 7.8196 |
| VisionTS++ | **0.8082** | **0.3164** | **5.2999** | *7.7330* |
| **ML Baselines** | | | | |
| DLinear | 0.8401 | 0.3388 | 5.5012 | **7.6941** |
| DeepAR | 0.8583 | *0.3183* | 5.5993 | 8.3161 |
| LightGBM | 0.8825 | 0.3425 | 5.7595 | 8.3442 |
| PatchTST | 0.8410 | 0.3359 | 5.5018 | 7.7037 |
| **Statistical Baselines** | | | | |
| AutoETS | 0.9284 | 0.3863 | 6.1033 | 8.4334 |
| Seasonal Naive | 1.0613 | 0.4610 | 6.9819 | 9.8980 |

*Table 26.* PM10 leaderboard — EEA DE

| model | MASE | CRPS | MAE | RMSE |
|---|---|---|---|---|
| **TSFMs** | | | | |
| Chronos-2 | 0.8343 | 0.3130 | 5.0470 | 9.1636 |
| Chronos-Bolt | 0.8445 | 0.3183 | 5.1055 | 9.2909 |
| Kairos | 0.9354 | 0.3616 | 5.6878 | 9.5998 |
| Moirai-1 | 0.8397 | 0.3182 | 5.0686 | 9.0318 |
| Moirai-2 | 0.8287 | *0.3048* | 4.9681 | 8.8505 |
| Sundial | 0.8407 | 0.3462 | 5.0761 | 8.8550 |
| TiRex | 0.8246 | 0.3084 | 4.9517 | 8.8209 |
| TimesFM-1.0 | 0.8601 | 0.3302 | 5.1837 | 9.0384 |
| TimesFM-2.0 | 0.8372 | 0.3166 | 5.0500 | 8.9691 |
| TimesFM-2.5 | *0.8187* | 0.3021 | 4.8999 | 8.7279 |
| VisionTS++ | 0.8177 | 0.3064 | *4.9071* | *8.7896* |
| **ML Baselines** | | | | |
| DLinear | 0.8303 | 0.3163 | 5.0137 | **8.7273** |
| DeepAR | 0.8602 | 0.3305 | 5.2019 | 9.1086 |
| LightGBM | 0.9093 | 0.3527 | 5.5419 | 9.4463 |
| PatchTST | **0.8155** | **0.3010** | **4.8939** | 8.8157 |
| **Statistical Baselines** | | | | |
| AutoETS | 0.9243 | 0.3816 | 5.6784 | 9.6087 |
| Seasonal Naive | 1.0899 | 0.4780 | 6.7188 | 11.5799 |

*Table 27.* PM2.5 leaderboard — EEA DE

| model | MASE | CRPS | MAE | RMSE |
|---|---|---|---|---|
| **TSFMs** | | | | |
| Chronos-2 | 0.8324 | 0.3375 | 3.3436 | 5.9400 |
| Chronos-Bolt | 0.8476 | 0.3465 | 3.4119 | 6.0666 |
| Kairos | 0.9508 | 0.4088 | 3.8665 | 6.3014 |
| Moirai-1 | 0.8391 | 0.3489 | 3.3854 | 5.8574 |
| Moirai-2 | 0.8243 | **0.3304** | 3.2947 | 5.6964 |
| Sundial | 0.8448 | 0.3826 | 3.4047 | 5.7638 |
| TiRex | *0.8206* | *0.3337* | *3.2833* | 5.6727 |
| TimesFM-1.0 | 0.8688 | 0.3679 | 3.5036 | 5.9506 |
| TimesFM-2.0 | 0.8330 | 0.3456 | 3.3427 | 5.7231 |
| TimesFM-2.5 | **0.8163** | 0.3307 | **3.2621** | 5.6238 |
| VisionTS++ | 0.8179 | 0.3344 | 3.2699 | *5.6720* |
| **ML Baselines** | | | | |
| DLinear | 0.8252 | 0.3424 | 3.3132 | **5.6214** |
| DeepAR | 0.8979 | 0.3747 | 3.6470 | 6.2293 |
| LightGBM | 0.9464 | 0.4412 | 3.9312 | 6.3485 |
| PatchTST | 0.8463 | 0.3375 | 3.3757 | 5.7457 |
| **Statistical Baselines** | | | | |
| AutoETS | 0.9290 | 0.4193 | 3.7812 | 6.2433 |
| Seasonal Naive | 1.1014 | 0.5335 | 4.5330 | 7.6529 |

## D.5. EEA-France

*Table 29.* CO leaderboard — EEA FR

| model | MASE | CRPS | MAE | RMSE |
|---|---|---|---|---|
| **TSFMs** | | | | |
| Chronos-2 | 0.8230 | 0.1978 | 0.0571 | 0.0954 |
| Chronos-Bolt | 0.8259 | 0.1990 | 0.0571 | 0.0994 |
| Kairos | 0.9988 | 0.2401 | 0.0676 | 0.1044 |
| Moirai-1 | 0.8493 | 0.2062 | 0.0584 | 0.0955 |
| Moirai-2 | 0.8176 | *0.1963* | 0.0563 | 0.0928 |
| Sundial | 0.8143 | 0.2127 | 0.0563 | *0.0920* |
| TiRex | *0.8058* | 0.1964 | 0.0561 | 0.0931 |
| TimesFM-1.0 | 0.8310 | 0.2039 | 0.0577 | 0.0950 |
| TimesFM-2.0 | 0.8162 | 0.1979 | *0.0561* | 0.0928 |
| TimesFM-2.5 | 0.8053 | 0.1936 | 0.0554 | 0.0921 |
| VisionTS++ | **0.7966** | **0.1917** | **0.0548** | **0.0908** |
| **ML Baselines** | | | | |
| DLinear | 0.8224 | 0.2050 | 0.0567 | 0.0909 |
| DeepAR | 0.8968 | 0.2189 | 0.0608 | 0.0954 |
| LightGBM | 0.9137 | 0.2236 | 0.0626 | 0.0966 |
| PatchTST | 0.8456 | 0.2062 | 0.0577 | 0.0924 |
| **Statistical Baselines** | | | | |
| AutoETS | 0.9259 | 0.2296 | 0.0639 | 0.0997 |
| Seasonal Naive | 1.0679 | 0.2772 | 0.0745 | 0.1206 |

*Table 28.* SO2 leaderboard — EEA DE

| model | MASE | CRPS | MAE | RMSE |
|---|---|---|---|---|
| **TSFMs** | | | | |
| Chronos-2 | 1.1408 | 0.3855 | 0.8246 | 2.5037 |
| Chronos-Bolt | 1.1519 | 0.3872 | 0.8230 | 2.5826 |
| Kairos | 1.2037 | 0.5267 | 0.8707 | 2.5357 |
| Moirai-1 | 1.1489 | 0.4005 | 0.8127 | 2.5060 |
| Moirai-2 | *1.1236* | 0.3640 | *0.8014* | 2.4900 |
| Sundial | 1.1461 | 0.4773 | 0.8518 | **2.4232** |
| TiRex | **1.1158** | *0.3690* | **0.7872** | *2.4648* |
| TimesFM-1.0 | 1.4292 | 0.3899 | 0.8163 | 2.4760 |
| TimesFM-2.0 | 1.2457 | 0.4151 | 0.8065 | 2.4744 |
| TimesFM-2.5 | 1.1255 | **0.3612** | 0.7931 | 2.4648 |
| VisionTS++ | 1.1184 | 0.3820 | 0.8090 | 2.4838 |
| **ML Baselines** | | | | |
| DLinear | 3.1786 | 0.5793 | 0.9505 | 2.4537 |
| DeepAR | 1.4626 | 0.4529 | 0.8335 | 2.5126 |
| LightGBM | 4.3892 | 0.4768 | 0.8417 | 2.4860 |
| PatchTST | 1.3691 | 0.4049 | 0.8104 | 2.4971 |
| **Statistical Baselines** | | | | |
| AutoETS | 1.3345 | 0.7793 | 1.1364 | 2.6697 |
| Seasonal Naive | 1.4236 | 1.0430 | 1.1851 | 3.2927 |

*Table 30.* NO2 leaderboard — EEA FR

| model | MASE | CRPS | MAE | RMSE |
|---|---|---|---|---|
| **TSFMs** | | | | |
| Chronos-2 | 0.8275 | 0.3636 | 5.3837 | 8.2704 |
| Chronos-Bolt | 0.8176 | 0.3629 | 5.3119 | 8.1386 |
| Kairos | 1.1078 | 0.4820 | 7.2358 | 10.2930 |
| Moirai-1 | 0.8531 | 0.3777 | 5.5467 | 8.3977 |
| Moirai-2 | 0.8313 | 0.3633 | 5.4086 | 8.2653 |
| Sundial | 0.8350 | 0.4102 | 5.4487 | 8.2165 |
| TiRex | 0.8293 | 0.3667 | 5.4086 | 8.3011 |
| TimesFM-1.0 | 0.8384 | 0.3786 | 5.4695 | 8.3687 |
| TimesFM-2.0 | 0.8255 | 0.3599 | 5.3664 | 8.2945 |
| TimesFM-2.5 | *0.8208* | 0.3569 | *5.3355* | 8.1879 |
| VisionTS++ | **0.8131** | **0.3560** | **5.2793** | *8.1269* |
| **ML Baselines** | | | | |
| DLinear | 0.8367 | 0.3884 | 5.4310 | **8.0152** |
| DeepAR | 0.8484 | *0.3584* | 5.4845 | 8.4766 |
| LightGBM | 0.8789 | 0.4290 | 5.7480 | 8.3958 |
| PatchTST | 0.8357 | 0.3777 | 5.4206 | 8.0296 |
| **Statistical Baselines** | | | | |
| AutoETS | 0.9456 | 0.4554 | 6.1492 | 8.7922 |
| Seasonal Naive | 1.0544 | 0.5358 | 6.8872 | 10.2616 |

*Table 31.* Ozone leaderboard — EEA FR

| model | MASE | CRPS | MAE | RMSE |
|---|---|---|---|---|
| **TSFMs** | | | | |
| Chronos-2 | 0.8146 | *0.2062* | 12.0841 | 16.3168 |
| Chronos-Bolt | 0.8119 | 0.2064 | 12.0452 | 16.2661 |
| Kairos | 1.2111 | 0.3034 | 17.9576 | 23.5162 |
| Moirai-1 | 0.8450 | 0.2158 | 12.5607 | 16.6852 |
| Moirai-2 | 0.8332 | 0.2137 | 12.3981 | 16.6311 |
| Sundial | 0.8170 | 0.2298 | 12.1212 | 16.0334 |
| TiRex | **0.7951** | **0.2044** | **11.7964** | 15.9004 |
| TimesFM-1.0 | 0.8733 | 0.2297 | 13.0399 | 17.2936 |
| TimesFM-2.0 | 0.8154 | 0.2123 | 12.0961 | 16.1078 |
| TimesFM-2.5 | 0.8012 | 0.2066 | 11.8760 | **15.8890** |
| VisionTS++ | *0.8055* | 0.2047 | *11.9388* | *16.0248* |
| **ML Baselines** | | | | |
| DLinear | 0.8365 | 0.2299 | 12.3501 | 16.2921 |
| DeepAR | 1.1036 | 0.3080 | 16.0115 | 20.6528 |
| LightGBM | 0.8953 | 0.2360 | 13.2572 | 16.9690 |
| PatchTST | 0.8334 | 0.2227 | 12.3068 | 16.1720 |
| **Statistical Baselines** | | | | |
| AutoETS | 0.9205 | 0.2409 | 13.6965 | 17.6835 |
| Seasonal Naive | 1.0497 | 0.2836 | 15.7446 | 20.7604 |

*Table 33.* PM2.5 leaderboard — EEA FR

| model | MASE | CRPS | MAE | RMSE |
|---|---|---|---|---|
| **TSFMs** | | | | |
| Chronos-2 | 0.8328 | 0.3404 | 3.2385 | 5.4543 |
| Chronos-Bolt | 0.8415 | 0.3453 | 3.2647 | 5.5422 |
| Kairos | 0.9590 | 0.4093 | 3.8036 | 6.1093 |
| Moirai-1 | 0.8430 | 0.3508 | 3.3001 | 5.5336 |
| Moirai-2 | 0.8268 | *0.3344* | 3.1962 | 5.3406 |
| Sundial | 0.8452 | 0.3819 | 3.2904 | 5.4066 |
| TiRex | 0.8207 | 0.3380 | 3.1867 | 5.3400 |
| TimesFM-1.0 | 0.8572 | 0.3593 | 3.3354 | 5.5124 |
| TimesFM-2.0 | 0.8389 | 0.3464 | 3.2634 | 5.4468 |
| TimesFM-2.5 | *0.8195* | 0.3331 | *3.1692* | *5.3046* |
| VisionTS++ | **0.8163** | 0.3358 | 3.1556 | 5.2660 |
| **ML Baselines** | | | | |
| DLinear | 0.8386 | 0.3463 | 3.2626 | 5.3056 |
| DeepAR | 0.8494 | 0.3457 | 3.2760 | 5.4698 |
| LightGBM | 0.9460 | 0.3584 | 3.5776 | 5.9298 |
| PatchTST | 0.8189 | **0.3304** | **3.1556** | **5.2533** |
| **Statistical Baselines** | | | | |
| AutoETS | 0.9203 | 0.4077 | 3.6309 | 5.8095 |
| Seasonal Naive | 1.0877 | 0.5087 | 4.2827 | 6.8934 |

*Table 32.* PM10 leaderboard — EEA FR

| model | MASE | CRPS | MAE | RMSE |
|---|---|---|---|---|
| **TSFMs** | | | | |
| Chronos-2 | 0.8385 | 0.3202 | 5.3807 | 8.9583 |
| Chronos-Bolt | 0.8409 | 0.3247 | 5.3937 | 9.0215 |
| Kairos | 0.9506 | 0.3699 | 6.1405 | 9.7315 |
| Moirai-1 | 0.8481 | 0.3257 | 5.4427 | 8.9578 |
| Moirai-2 | 0.8368 | *0.3140* | 5.3384 | 8.8236 |
| Sundial | 0.8434 | 0.3555 | 5.4162 | 8.8337 |
| TiRex | 0.8293 | 0.3191 | 5.3070 | 8.7965 |
| TimesFM-1.0 | 0.8575 | 0.3311 | 5.4873 | 8.9984 |
| TimesFM-2.0 | 0.8452 | 0.3234 | 5.4150 | 8.9688 |
| TimesFM-2.5 | *0.8268* | **0.3109** | 5.2671 | 8.7433 |
| VisionTS++ | **0.8201** | 0.3128 | **5.2316** | 8.6742 |
| **ML Baselines** | | | | |
| DLinear | 0.8430 | 0.3370 | 5.4519 | *8.7244* |
| DeepAR | 0.8672 | 0.3161 | 5.4928 | 9.2380 |
| LightGBM | 0.9190 | 0.3347 | 5.7991 | 9.5348 |
| PatchTST | 0.8264 | 0.3246 | *5.2885* | **8.6473** |
| **Statistical Baselines** | | | | |
| AutoETS | 0.9247 | 0.3824 | 6.0361 | 9.5329 |
| Seasonal Naive | 1.0795 | 0.4722 | 7.0526 | 11.4082 |

*Table 34.* SO2 leaderboard — EEA FR

| model | MASE | CRPS | MAE | RMSE |
|---|---|---|---|---|
| **TSFMs** | | | | |
| Chronos-2 | 1.0841 | 0.6271 | 1.4486 | 5.4111 |
| Chronos-Bolt | 1.0895 | 0.6357 | 1.4504 | 5.5090 |
| Kairos | 1.1456 | 0.8756 | 1.5378 | 5.3951 |
| Moirai-1 | 1.0801 | *0.5783* | 1.4103 | 5.2885 |
| Moirai-2 | 1.0719 | 0.5813 | 1.4000 | 5.2781 |
| Sundial | 1.0877 | 0.9501 | 1.5083 | **5.2055** |
| TiRex | **1.0594** | 0.5987 | **1.3797** | 5.2516 |
| TimesFM-1.0 | 1.0877 | 0.6174 | 1.4257 | 5.2673 |
| TimesFM-2.0 | 1.0798 | 0.6054 | 1.4012 | 5.2571 |
| TimesFM-2.5 | *1.0687* | 0.5512 | 1.3831 | *5.2422* |
| VisionTS++ | 1.0627 | 0.6070 | 1.4137 | 5.2672 |
| **ML Baselines** | | | | |
| DLinear | 1.1879 | 0.9633 | 1.6356 | 5.2230 |
| DeepAR | 1.0962 | **0.5242** | 1.4141 | 5.3753 |
| LightGBM | 1.1133 | 0.8002 | 1.4508 | 5.2480 |
| PatchTST | 1.0905 | 0.6690 | *1.3976* | 5.3111 |
| **Statistical Baselines** | | | | |
| AutoETS | 1.2921 | 2.0221 | 2.1085 | 5.7417 |
| Seasonal Naive | 1.3894 | 2.6173 | 2.1497 | 7.1615 |

## D.6. EPA

*Table 35.* CO leaderboard — EPA

| model | MASE | CRPS | MAE | RMSE |
|---|---|---|---|---|
| **TSFMs** | | | | |
| Chronos-2 | 0.8876 | 0.4234 | 0.0924 | 0.1540 |
| Chronos-Bolt | 0.8950 | 0.4590 | 0.0930 | 0.1551 |
| Kairos | 1.1074 | 0.5472 | 0.1186 | 0.1837 |
| Moirai-1 | 0.9165 | 0.4215 | 0.0955 | 0.1566 |
| Moirai-2 | 0.8918 | **0.4034** | 0.0927 | 0.1530 |
| Sundial | 0.9020 | 0.5943 | 0.0941 | 0.1518 |
| TiRex | 0.8783 | 0.4389 | *0.0918* | 0.1531 |
| TimesFM-1.0 | 0.9270 | 0.5742 | 0.0961 | 0.1566 |
| TimesFM-2.0 | 0.9012 | 0.5556 | 0.0935 | 0.1545 |
| TimesFM-2.5 | *0.8840* | *0.4194* | 0.0916 | 0.1513 |
| VisionTS++ | **0.8694** | 0.4490 | **0.0904** | **0.1501** |
| **ML Baselines** | | | | |
| DLinear | 1.1380 | 0.6602 | 0.0958 | *0.1515* |
| DeepAR | 1.3739 | 0.4077 | 0.1003 | 0.1606 |
| LightGBM | 1.0428 | 0.5774 | 0.0965 | 0.1567 |
| PatchTST | 0.9606 | 0.4638 | 0.0936 | 0.1551 |
| **Statistical Baselines** | | | | |
| AutoETS | 1.0302 | 0.6398 | 0.1094 | 0.1670 |
| Seasonal Naive | 1.1079 | 0.7980 | 0.1195 | 0.1996 |

*Table 37.* Ozone leaderboard — EPA

| model | MASE | CRPS | MAE | RMSE |
|---|---|---|---|---|
| **TSFMs** | | | | |
| Chronos-2 | 0.8380 | 0.1851 | 12.2629 | 16.6584 |
| Chronos-Bolt | 0.8291 | 0.1847 | 12.1424 | 16.4878 |
| Kairos | 1.4682 | 0.3221 | 21.3094 | 27.7594 |
| Moirai-1 | 0.8718 | 0.1984 | 12.7875 | 17.1063 |
| Moirai-2 | 0.8546 | 0.1941 | 12.5470 | 16.8949 |
| Sundial | 0.8285 | 0.2080 | 12.1178 | *16.1548* |
| TiRex | **0.8184** | **0.1826** | 11.9921 | 16.2554 |
| TimesFM-1.0 | 0.8909 | 0.2158 | 13.1105 | 17.5651 |
| TimesFM-2.0 | 0.8369 | 0.1949 | 12.2496 | 16.4241 |
| TimesFM-2.5 | *0.8237* | *0.1849* | *12.0543* | 16.2240 |
| VisionTS++ | 0.8270 | 0.1863 | 12.1034 | 16.3623 |
| **ML Baselines** | | | | |
| DLinear | 0.8399 | 0.2005 | 12.2227 | 16.1133 |
| DeepAR | 0.8981 | 0.2354 | 13.0375 | 17.2487 |
| LightGBM | 0.8539 | 0.2097 | 12.4312 | 16.3009 |
| PatchTST | 0.8208 | 0.1981 | **11.9470** | **15.7365** |
| **Statistical Baselines** | | | | |
| AutoETS | 0.9526 | 0.2184 | 13.9780 | 18.1409 |
| Seasonal Naive | 1.0477 | 0.2486 | 15.4815 | 20.6695 |

*Table 36.* NO2 leaderboard — EPA

| model | MASE | CRPS | MAE | RMSE |
|---|---|---|---|---|
| **TSFMs** | | | | |
| Chronos-2 | 0.8219 | 0.4729 | 6.3948 | 9.6644 |
| Chronos-Bolt | 0.8195 | 0.4806 | 6.3767 | 9.5964 |
| Kairos | 1.0933 | 0.6577 | 8.4913 | 11.8802 |
| Moirai-1 | 0.8351 | 0.5070 | 6.5003 | 9.7437 |
| Moirai-2 | 0.8207 | 0.4768 | 6.3987 | 9.6603 |
| Sundial | 0.8345 | 0.5839 | 6.5398 | 9.6261 |
| TiRex | *0.8184* | *0.4707* | 6.3913 | 9.6839 |
| TimesFM-1.0 | 0.8374 | 0.5149 | 6.5608 | 9.8290 |
| TimesFM-2.0 | 0.8250 | 0.4863 | 6.4367 | 9.7710 |
| TimesFM-2.5 | 0.8134 | 0.4632 | 6.3313 | 9.5473 |
| VisionTS++ | **0.8028** | 0.4775 | **6.2476** | 9.4596 |
| **ML Baselines** | | | | |
| DLinear | 0.8330 | 0.5587 | 6.4987 | *9.4786* |
| DeepAR | 0.8551 | **0.4241** | 6.5006 | 9.8773 |
| LightGBM | 0.8763 | 0.6080 | 6.8518 | 9.7412 |
| PatchTST | 0.8190 | 0.4999 | *6.3553* | **9.3660** |
| **Statistical Baselines** | | | | |
| AutoETS | 0.9506 | 0.6841 | 7.4490 | 10.4995 |
| Seasonal Naive | 1.0615 | 0.8330 | 8.4073 | 12.4848 |

*Table 38.* PM10 leaderboard — EPA

| model | MASE | CRPS | MAE | RMSE |
|---|---|---|---|---|
| **TSFMs** | | | | |
| Chronos-2 | 0.8628 | 0.3777 | 10.1982 | 27.9851 |
| Chronos-Bolt | 0.8668 | 0.3813 | 10.3085 | 29.0075 |
| Kairos | 0.9902 | 0.4428 | 11.8381 | 29.2064 |
| Moirai-1 | 0.8765 | 0.3827 | 10.3722 | 28.0290 |
| Moirai-2 | 0.8642 | *0.3708* | 10.1868 | 27.8411 |
| Sundial | 0.8736 | 0.4338 | 10.4963 | **27.5656** |
| TiRex | 0.8556 | 0.3747 | *10.0847* | *27.6752* |
| TimesFM-1.0 | 0.8754 | 0.3802 | 10.3203 | 27.8795 |
| TimesFM-2.0 | 0.8712 | 0.3733 | 10.2584 | 27.9821 |
| TimesFM-2.5 | *0.8567* | **0.3644** | 10.0779 | 27.7150 |
| VisionTS++ | **0.8476** | 0.3698 | **10.0027** | 27.6283 |
| **ML Baselines** | | | | |
| DLinear | 0.8938 | 0.4170 | 11.1507 | 28.2917 |
| DeepAR | 0.8966 | 0.3865 | 10.4667 | 28.1655 |
| LightGBM | 0.9009 | 0.3937 | 10.5341 | 28.1171 |
| PatchTST | 0.8570 | 0.3716 | 10.1313 | 27.8679 |
| **Statistical Baselines** | | | | |
| AutoETS | 1.0000 | 0.5669 | 13.1126 | 30.6961 |
| Seasonal Naive | 1.1234 | 0.7049 | 14.0309 | 37.1597 |

*Table 39.* PM2.5 leaderboard — EPA

| model | MASE | CRPS | MAE | RMSE |
|---|---|---|---|---|
| **TSFMs** | | | | |
| Chronos-2 | 0.8239 | 0.3574 | 3.0071 | 5.6841 |
| Chronos-Bolt | 0.8319 | 0.3623 | 3.0459 | 6.0134 |
| Kairos | 0.9258 | 0.4147 | 3.4086 | 5.9857 |
| Moirai-1 | 0.8321 | 0.3649 | 3.0339 | 5.6331 |
| Moirai-2 | 0.8164 | _0.3494_ | 2.9580 | 5.5068 |
| Sundial | 0.8258 | 0.3960 | 3.0149 | 5.4862 |
| TiRex | _0.8071_ | 0.3514 | _2.9328_ | 5.4541 |
| TimesFM-1.0 | 0.8379 | 0.3669 | 3.0485 | 5.5921 |
| TimesFM-2.0 | 0.8287 | 0.3609 | 3.0221 | 5.5883 |
| TimesFM-2.5 | _0.8072_ | **0.3462** | _2.9242_ | _5.4421_ |
| VisionTS++ | **0.8045** | _0.3500_ | **2.9182** | **5.4345** |
| **ML Baselines** | | | | |
| DLinear | 0.8338 | 0.3634 | 3.0381 | 5.4803 |
| DeepAR | 0.8302 | 0.3587 | 3.0094 | 5.5858 |
| LightGBM | 0.8590 | 0.3886 | 3.1265 | 5.6449 |
| PatchTST | 0.8184 | 0.3634 | 2.9643 | _5.4390_ |
| **Statistical Baselines** | | | | |
| AutoETS | 0.9187 | 0.4386 | 3.4324 | 6.0979 |
| Seasonal Naive | 1.0868 | 0.5546 | 4.0498 | 7.3536 |

## D.7. SINAICA

*Table 41.* CO leaderboard — SINAICA

| model | MASE | CRPS | MAE | RMSE |
|---|---|---|---|---|
| **TSFMs** | | | | |
| Chronos-2 | 1.6459 | 0.1534 | 0.4190 | 11.3693 |
| Chronos-Bolt | 1.6518 | 0.9410 | 0.6650 | 14.8405 |
| Kairos | 2.0228 | 0.4077 | 0.5559 | 11.5254 |
| Moirai-1 | 1.6864 | 0.1517 | 0.4240 | 11.3715 |
| Moirai-2 | 1.6456 | _0.1486_ | 0.4164 | 11.3628 |
| Sundial | 1.6552 | 0.2156 | 0.4415 | **11.3511** |
| TiRex | _1.6403_ | 0.1628 | _0.4151_ | 11.3623 |
| TimesFM-1.0 | 1.6776 | 0.1766 | 0.4314 | 11.3675 |
| TimesFM-2.0 | 1.6555 | 0.1542 | 0.4232 | 11.3645 |
| TimesFM-2.5 | _1.6387_ | **0.1441** | _0.4114_ | _11.3532_ |
| VisionTS++ | **1.6296** | _0.1480_ | **0.4096** | _11.3598_ |
| **ML Baselines** | | | | |
| DLinear | 1.7215 | 0.5229 | 0.6342 | 12.0907 |
| DeepAR | 1.8409 | 0.3589 | 0.5240 | 11.4274 |
| LightGBM | 1.7096 | 0.3526 | 0.5807 | 12.0757 |
| PatchTST | 1.6721 | 0.2506 | 0.4678 | 11.3776 |
| **Statistical Baselines** | | | | |
| AutoETS | 1.7730 | 1.0845 | 0.7294 | 12.1412 |
| Seasonal Naive | 1.8727 | 1.3228 | 0.6921 | 15.9470 |

*Table 40.* SO2 leaderboard — EPA

| model | MASE | CRPS | MAE | RMSE |
|---|---|---|---|---|
| **TSFMs** | | | | |
| Chronos-2 | 1.2588 | 2.3582 | 1.3193 | 4.4735 |
| Chronos-Bolt | 1.2631 | 2.6726 | 1.3060 | 4.4660 |
| Kairos | 1.3632 | 4.2472 | 1.4510 | 4.5258 |
| Moirai-1 | 1.2703 | 2.1797 | 1.2996 | 4.4258 |
| Moirai-2 | 1.2544 | _1.8346_ | _1.2766_ | 4.3716 |
| Sundial | 1.2805 | 4.8651 | 1.3758 | **4.2742** |
| TiRex | **1.2411** | 2.2390 | **1.2516** | _4.3317_ |
| TimesFM-1.0 | 1.4186 | 2.9853 | 1.3194 | 4.3768 |
| TimesFM-2.0 | 1.3182 | 2.7053 | 1.2922 | 4.3542 |
| TimesFM-2.5 | _1.2488_ | 1.8708 | _1.2691_ | _4.3334_ |
| VisionTS++ | _1.2463_ | 2.5548 | 1.2969 | 4.3692 |
| **ML Baselines** | | | | |
| DLinear | 1.4244 | 3.0468 | 1.5137 | 4.5582 |
| DeepAR | 1.3642 | **0.7531** | 1.3597 | 4.5083 |
| LightGBM | 1.3163 | 3.2585 | 1.3360 | 4.4334 |
| PatchTST | 1.3065 | _1.2619_ | 1.3155 | 4.5736 |
| **Statistical Baselines** | | | | |
| AutoETS | 1.4663 | 11.4132 | 1.8435 | 4.6999 |
| Seasonal Naive | 1.5381 | 16.0548 | 1.8590 | 5.7495 |

*Table 42.* NO2 leaderboard — SINAICA

| model | MASE | CRPS | MAE | RMSE |
|---|---|---|---|---|
| **TSFMs** | | | | |
| Chronos-2 | 0.8690 | 0.2712 | 9.6182 | 14.7003 |
| Chronos-Bolt | _0.8669_ | 0.2734 | 9.5991 | 14.6379 |
| Kairos | 1.1896 | 0.3745 | 12.6191 | 17.8018 |
| Moirai-1 | 0.8812 | 0.2781 | 9.7046 | 14.6250 |
| Moirai-2 | 0.8685 | _0.2704_ | 9.5532 | 14.5262 |
| Sundial | 0.8800 | 0.2997 | 9.6786 | _14.4054_ |
| TiRex | 0.8723 | 0.2741 | 9.6286 | 14.6021 |
| TimesFM-1.0 | 0.8953 | 0.2864 | 9.8791 | 14.8711 |
| TimesFM-2.0 | 0.8718 | 0.2718 | _9.5517_ | 14.4977 |
| TimesFM-2.5 | _0.8683_ | _0.2699_ | _9.5247_ | 14.4222 |
| VisionTS++ | **0.8508** | **0.2660** | **9.3498** | _14.2366_ |
| **ML Baselines** | | | | |
| DLinear | 0.9317 | 0.2989 | 10.2054 | 14.8442 |
| DeepAR | 0.9074 | 0.2959 | 9.8621 | 14.5161 |
| LightGBM | 0.9113 | 0.2846 | 9.9901 | 15.0076 |
| PatchTST | 0.8993 | 0.2882 | 9.7980 | **14.2044** |
| **Statistical Baselines** | | | | |
| AutoETS | 0.9776 | 0.3215 | 10.8688 | 15.7756 |
| Seasonal Naive | 1.0652 | 0.3690 | 11.9852 | 18.1850 |

*Table 43.* Ozone leaderboard — SINAICA

| model | MASE | CRPS | MAE | RMSE |
|---|---|---|---|---|
| **TSFMs** | | | | |
| Chronos-2 | *0.9148* | 0.2596 | *15.5821* | 23.9814 |
| Chronos-Bolt | 0.9280 | 0.2616 | 15.8567 | 24.7865 |
| Kairos | 1.7499 | 0.4667 | 29.4913 | 42.0203 |
| Moirai-1 | 0.9664 | 0.2784 | 16.5288 | 25.2773 |
| Moirai-2 | 0.9261 | 0.2602 | 15.7702 | 24.2713 |
| Sundial | 0.9369 | 0.2897 | 16.0107 | 24.4242 |
| TiRex | 0.9205 | *0.2590* | 15.7612 | 24.4908 |
| TimesFM-1.0 | 0.9849 | 0.2776 | 16.7733 | 25.6076 |
| TimesFM-2.0 | 0.9520 | 0.2708 | 16.2341 | 25.0203 |
| TimesFM-2.5 | 0.9214 | 0.2618 | 15.6947 | *24.1613* |
| VisionTS++ | 0.9126 | **0.2561** | 15.5572 | 24.4462 |
| **ML Baselines** | | | | |
| DLinear | 0.9437 | 0.2689 | 16.0446 | 24.4160 |
| DeepAR | 0.9585 | 0.2786 | 16.2756 | 24.9166 |
| LightGBM | 0.9979 | 0.2908 | 17.0597 | 25.4262 |
| PatchTST | **0.9095** | 0.2589 | **15.4667** | **23.4825** |
| **Statistical Baselines** | | | | |
| AutoETS | 1.0582 | 0.3151 | 18.1458 | 26.4147 |
| Seasonal Naive | 1.0679 | 0.3324 | 18.4489 | 28.8281 |

*Table 45.* PM2.5 leaderboard — SINAICA

| model | MASE | CRPS | MAE | RMSE |
|---|---|---|---|---|
| **TSFMs** | | | | |
| Chronos-2 | 0.8906 | 0.6279 | 8.4020 | 22.7269 |
| Chronos-Bolt | 0.8899 | 0.5048 | 8.4168 | 22.3098 |
| Kairos | 1.1324 | 1.0800 | 10.5028 | 23.6540 |
| Moirai-1 | 0.9005 | 0.4655 | 8.6445 | 24.2264 |
| Moirai-2 | 0.8936 | 0.4425 | 8.4412 | 22.9741 |
| Sundial | 0.9017 | 0.9137 | 8.6400 | *22.3246* |
| TiRex | 0.8839 | 0.5195 | *8.2597* | 22.6320 |
| TimesFM-1.0 | 0.9192 | 1.0721 | 9.0153 | 23.4278 |
| TimesFM-2.0 | 0.8996 | 0.7025 | 8.5679 | 22.6204 |
| TimesFM-2.5 | *0.8861* | **0.4317** | 8.2579 | 22.4919 |
| VisionTS++ | **0.8713** | *0.4431* | **8.1691** | **22.0670** |
| **ML Baselines** | | | | |
| DLinear | 0.9109 | 1.2751 | 9.1799 | 22.4266 |
| DeepAR | 0.9390 | 1.8948 | 9.5727 | 24.6575 |
| LightGBM | 0.9462 | 1.5072 | 9.5261 | 24.7793 |
| PatchTST | 0.8915 | 0.7974 | 8.8777 | 23.5038 |
| **Statistical Baselines** | | | | |
| AutoETS | 0.9850 | 1.1442 | 9.8096 | 23.5795 |
| Seasonal Naive | 1.1052 | 1.8821 | 10.8728 | 29.3552 |

*Table 46.* SO2 leaderboard — SINAICA

| model | MASE | CRPS | MAE | RMSE |
|---|---|---|---|---|
| **TSFMs** | | | | |
| Chronos-2 | 0.9246 | 0.1813 | 2.9853 | 7.3655 |
| Chronos-Bolt | 0.9344 | 0.1839 | 3.0252 | 7.4160 |
| Kairos | 1.0302 | 0.2018 | 3.2697 | 7.6795 |
| Moirai-1 | 0.9414 | 0.1834 | 3.0147 | 7.5182 |
| Moirai-2 | *0.9222* | 0.1795 | 2.9725 | 7.4098 |
| Sundial | 0.9383 | 0.2021 | 3.1036 | **7.2568** |
| TiRex | 0.9164 | 0.1783 | 2.9485 | 7.3934 |
| TimesFM-1.0 | 0.9511 | 0.1870 | 3.0541 | 7.5036 |
| TimesFM-2.0 | 0.9370 | 0.1816 | 2.9984 | 7.4918 |
| TimesFM-2.5 | 0.9251 | *0.1790* | 2.9733 | 7.4304 |
| VisionTS++ | **0.9116** | **0.1775** | **2.9349** | 7.3564 |
| **ML Baselines** | | | | |
| DLinear | 0.9927 | 0.2038 | 3.2747 | 7.2580 |
| DeepAR | 0.9955 | 0.1905 | 3.1216 | 7.6237 |
| LightGBM | 0.9586 | 0.1908 | 3.0566 | *7.3101* |
| PatchTST | 0.9277 | 0.1811 | *2.9670* | 7.4174 |
| **Statistical Baselines** | | | | |
| AutoETS | 1.0922 | 0.2530 | 3.8005 | 7.7580 |
| Seasonal Naive | 1.1596 | 0.3021 | 3.9892 | 9.3433 |

*Table 44.* PM10 leaderboard — SINAICA

| model | MASE | CRPS | MAE | RMSE |
|---|---|---|---|---|
| **TSFMs** | | | | |
| Chronos-2 | *0.8668* | 0.2898 | 21.1401 | 41.3729 |
| Chronos-Bolt | 0.8735 | 0.2956 | 21.3051 | 41.6046 |
| Kairos | 1.0590 | 0.3505 | 25.2880 | 44.6913 |
| Moirai-1 | 0.8914 | 0.2996 | 21.6874 | 40.9342 |
| Moirai-2 | 0.8698 | *0.2873* | 21.1115 | *40.3774* |
| Sundial | 0.8821 | 0.3284 | 21.6142 | 40.6053 |
| TiRex | 0.8700 | 0.2920 | *21.0696* | 40.4695 |
| TimesFM-1.0 | 0.8875 | 0.2973 | 21.6327 | 41.1887 |
| TimesFM-2.0 | 0.8744 | 0.2930 | 21.2906 | 40.8761 |
| TimesFM-2.5 | 0.8648 | **0.2844** | 20.9067 | 40.2202 |
| VisionTS++ | **0.8502** | 0.2846 | **20.6710** | **40.0188** |
| **ML Baselines** | | | | |
| DLinear | 0.8913 | 0.3118 | 22.1349 | 40.8321 |
| DeepAR | 0.8965 | 0.3126 | 22.2166 | 42.4870 |
| LightGBM | 0.9108 | 0.3215 | 22.7085 | 42.9164 |
| PatchTST | 0.8858 | 0.2934 | 21.7989 | 41.8590 |
| **Statistical Baselines** | | | | |
| AutoETS | 0.9814 | 0.3616 | 24.6224 | 43.8215 |
| Seasonal Naive | 1.0909 | 0.4343 | 27.8025 | 53.0974 |

# E. Dataset Statistics

Table 47 reports distributional statistics for each network-pollutant combination, averaged over sites.

*Table 47.* Dataset statistics per network and pollutant (mean over sites).

| network | pollutant | mean | median | std | skewness | p10 | p90 |
|---|---|---|---|---|---|---|---|
| AURN | CO (mg/m$^3$) | 0.19 | 0.15 | 0.14 | 4.56 | 0.09 | 0.31 |
| | NO$_2$ ($\mu$g/m$^3$) | 17.56 | 14.27 | 12.95 | 1.74 | 4.78 | 34.98 |
| | O$_3$ ($\mu$g/m$^3$) | 52.47 | 53.71 | 22.09 | 0.12 | 22.17 | 78.23 |
| | PM$_{10}$ ($\mu$g/m$^3$) | 13.41 | 11.17 | 9.59 | 4.15 | 4.78 | 24.51 |
| | PM$_{2.5}$ ($\mu$g/m$^3$) | 7.45 | 5.64 | 6.29 | 3.26 | 2.36 | 14.77 |
| | SO$_2$ ($\mu$g/m$^3$) | 1.12 | 0.58 | 3.54 | 23.05 | 0.21 | 1.72 |
| CNEMC | CO (mg/m$^3$) | 0.63 | 0.57 | 0.30 | 1.86 | 0.35 | 0.99 |
| | NO$_2$ ($\mu$g/m$^3$) | 22.52 | 17.98 | 16.10 | 1.64 | 6.98 | 44.71 |
| | O$_3$ ($\mu$g/m$^3$) | 67.54 | 62.17 | 40.79 | 0.70 | 18.23 | 123.87 |
| | PM$_{10}$ ($\mu$g/m$^3$) | 62.16 | 47.50 | 65.68 | 8.01 | 18.67 | 115.55 |
| | PM$_{2.5}$ ($\mu$g/m$^3$) | 31.09 | 23.43 | 28.94 | 4.80 | 8.22 | 62.24 |
| | SO$_2$ ($\mu$g/m$^3$) | 8.28 | 7.02 | 5.71 | 7.94 | 3.95 | 13.65 |
| CPCB | CO (mg/m$^3$) | 0.86 | 0.70 | 0.66 | 2.90 | 0.28 | 1.59 |
| | NO$_2$ ($\mu$g/m$^3$) | 24.66 | 19.45 | 19.78 | 3.41 | 7.85 | 47.62 |
| | O$_3$ ($\mu$g/m$^3$) | 29.18 | 21.71 | 25.15 | 2.21 | 6.93 | 61.54 |
| | PM$_{10}$ ($\mu$g/m$^3$) | 122.11 | 102.84 | 84.89 | 2.50 | 38.47 | 228.37 |
| | PM$_{2.5}$ ($\mu$g/m$^3$) | 57.07 | 42.51 | 50.16 | 3.86 | 14.81 | 117.65 |
| | SO$_2$ ($\mu$g/m$^3$) | 13.43 | 11.31 | 9.82 | 4.25 | 5.20 | 23.34 |
| EEA-DE | CO (mg/m$^3$) | 0.28 | 0.25 | 0.13 | 2.63 | 0.16 | 0.44 |
| | NO$_2$ ($\mu$g/m$^3$) | 15.07 | 12.77 | 9.76 | 1.68 | 5.13 | 28.16 |
| | O$_3$ ($\mu$g/m$^3$) | 53.46 | 52.76 | 27.79 | 0.35 | 16.75 | 89.97 |
| | PM$_{10}$ ($\mu$g/m$^3$) | 14.22 | 11.93 | 10.55 | 6.47 | 5.07 | 25.48 |
| | PM$_{2.5}$ ($\mu$g/m$^3$) | 8.92 | 6.95 | 7.55 | 5.38 | 2.71 | 17.32 |
| | SO$_2$ ($\mu$g/m$^3$) | 1.85 | 1.28 | 2.57 | 10.86 | 0.68 | 3.17 |
| EEA-FR | CO (mg/m$^3$) | 0.22 | 0.20 | 0.13 | 2.52 | 0.10 | 0.37 |
| | NO$_2$ ($\mu$g/m$^3$) | 14.24 | 11.43 | 10.54 | 2.01 | 4.27 | 28.03 |
| | O$_3$ ($\mu$g/m$^3$) | 56.43 | 56.72 | 26.34 | 0.22 | 20.86 | 89.42 |
| | PM$_{10}$ ($\mu$g/m$^3$) | 15.02 | 12.59 | 10.59 | 3.59 | 5.48 | 27.05 |
| | PM$_{2.5}$ ($\mu$g/m$^3$) | 8.63 | 6.63 | 7.31 | 3.55 | 2.59 | 16.82 |
| | SO$_2$ ($\mu$g/m$^3$) | 2.24 | 1.09 | 5.68 | 15.70 | 0.21 | 4.37 |
| EPA | CO (mg/m$^3$) | 0.33 | 0.27 | 0.20 | 3.00 | 0.15 | 0.57 |
| | NO$_2$ ($\mu$g/m$^3$) | 15.47 | 11.72 | 12.24 | 1.95 | 3.97 | 32.74 |
| | O$_3$ ($\mu$g/m$^3$) | 62.02 | 61.75 | 27.24 | 0.21 | 25.86 | 97.42 |
| | PM$_{10}$ ($\mu$g/m$^3$) | 22.23 | 16.51 | 31.79 | 14.12 | 6.02 | 40.67 |
| | PM$_{2.5}$ ($\mu$g/m$^3$) | 7.72 | 6.17 | 7.70 | 9.19 | 2.28 | 14.19 |
| | SO$_2$ ($\mu$g/m$^3$) | 2.58 | 1.24 | 5.39 | 9.88 | 0.23 | 5.71 |
| SINAICA | CO (mg/m$^3$) | 1.41 | 1.11 | 9.67 | 23.10 | 0.50 | 2.20 |
| | NO$_2$ ($\mu$g/m$^3$) | 29.82 | 24.11 | 20.45 | 1.65 | 10.61 | 57.23 |
| | O$_3$ ($\mu$g/m$^3$) | 52.47 | 44.97 | 42.75 | 4.25 | 11.47 | 100.21 |
| | PM$_{10}$ ($\mu$g/m$^3$) | 61.20 | 48.45 | 54.16 | 4.69 | 23.04 | 104.48 |
| | PM$_{2.5}$ ($\mu$g/m$^3$) | 22.50 | 15.83 | 34.27 | 5.52 | 5.96 | 37.66 |
| | SO$_2$ ($\mu$g/m$^3$) | 12.20 | 10.06 | 11.72 | 7.19 | 6.33 | 18.32 |

## F. Aggregated Results by Network and Pollutant

Table 48 and Table 49 disaggregate the overall leaderboard by network and by pollutant respectively, both normalized by Seasonal Naive. These tables reveal where relative model rankings remain stable and where task difficulty varies most, complementing the per-pollutant detail in Appendix D.

*Table 48.* Normalized MASE per dataset

| model | AURN | CNEMC | CPCB | EEA DE | EEA FR | EPA | SINAICA |
|---|---|---|---|---|---|---|---|
| **TSFMs** | | | | | | | |
| Chronos-2 | 0.7774 | 0.7854 | 0.8163 | 0.7779 | 0.7759 | 0.7886 | 0.8302 |
| Chronos-Bolt | 0.7795 | 0.7915 | 0.8303 | 0.7818 | 0.7769 | 0.7904 | 0.8347 |
| Kairos | 0.8721 | 1.0057 | 1.2579 | 0.9385 | 0.9471 | 0.9975 | 1.1117 |
| Moirai-1 | 0.7855 | 0.7961 | 0.8542 | 0.7934 | 0.7904 | 0.8043 | 0.8514 |
| Moirai-2 | 0.7687 | 0.7805 | 0.8181 | 0.7788 | 0.7754 | 0.7899 | 0.8321 |
| Sundial | 0.7685 | 0.7920 | 0.8265 | 0.7828 | 0.7791 | 0.7961 | 0.8414 |
| TiRex | 0.7582 | *0.7759* | 0.8083 | 0.7669 | 0.7638 | 0.7780 | 0.8291 |
| TimesFM-1.0 | 0.7896 | 0.8161 | 0.8822 | 0.8413 | 0.7944 | 0.8309 | 0.8579 |
| TimesFM-2.0 | 0.7711 | 0.7916 | 0.8369 | 0.7939 | 0.7759 | 0.8013 | 0.8409 |
| TimesFM-2.5 | *0.7592* | 0.7739 | *0.8105* | *0.7674* | *0.7642* | *0.7801* | *0.8292* |
| VisionTS++ | **0.7563** | **0.7702** | **0.8073** | **0.7642** | **0.7601** | **0.7749** | **0.8186** |
| **ML Baselines** | | | | | | | |
| DLinear | 0.7874 | 0.8238 | 1.1438 | 1.0870 | 0.7974 | 0.8561 | 0.8683 |
| DeepAR | 0.7860 | 0.8432 | 1.3517 | 0.8584 | 0.8414 | 0.8927 | 0.8881 |
| LightGBM | 0.8273 | 0.8599 | 0.9991 | 1.3366 | 0.8421 | 0.8398 | 0.8741 |
| PatchTST | 0.7605 | 0.7874 | 1.0544 | 0.8176 | 0.7803 | 0.8014 | 0.8403 |
| **Statistical Baselines** | | | | | | | |
| AutoETS | 0.8707 | 0.8865 | 0.9066 | 0.8807 | 0.8812 | 0.9071 | 0.9329 |
| Seasonal Naive | 1.0000 | 1.0000 | 1.0000 | 1.0000 | 1.0000 | 1.0000 | 1.0000 |

*Table 49.* Normalized MASE per pollutant

| model | CO | NO2 | Ozone | PM10 | PM2.5 | SO2 |
|---|---|---|---|---|---|---|
| **TSFMs** | | | | | | |
| Chronos-2 | 0.8015 | 0.7889 | 0.8016 | 0.7770 | 0.7708 | 0.8076 |
| Chronos-Bolt | 0.8077 | *0.7859* | 0.8052 | 0.7842 | 0.7801 | 0.8123 |
| Kairos | 0.9856 | 1.0424 | 1.3734 | 0.9027 | 0.9045 | 0.8795 |
| Moirai-1 | 0.8256 | 0.8054 | 0.8416 | 0.7876 | 0.7785 | 0.8160 |
| Moirai-2 | 0.7973 | 0.7883 | 0.8172 | 0.7737 | 0.7643 | 0.8012 |
| Sundial | 0.8015 | 0.7981 | 0.8077 | 0.7830 | 0.7770 | 0.8097 |
| TiRex | 0.7883 | 0.7883 | **0.7882** | *0.7692* | *0.7589* | **0.7932** |
| TimesFM-1.0 | 0.8208 | 0.8126 | 0.8733 | 0.7976 | 0.7935 | 0.8605 |
| TimesFM-2.0 | 0.8037 | 0.7931 | 0.8152 | 0.7813 | 0.7729 | 0.8265 |
| TimesFM-2.5 | *0.7897* | 0.7828 | 0.7940 | 0.7665 | 0.7574 | *0.7986* |
| VisionTS++ | **0.7825** | **0.7736** | *0.7954* | **0.7602** | **0.7536** | 0.7949 |
| **ML Baselines** | | | | | | |
| DLinear | 0.8693 | 0.8602 | 0.8788 | 0.8310 | 0.7881 | 1.0351 |
| DeepAR | 0.9164 | 0.9257 | 0.9669 | 0.8345 | 0.8071 | 0.9188 |
| LightGBM | 0.8910 | 0.8720 | 0.8804 | 0.8626 | 0.8470 | 1.0325 |
| PatchTST | 0.8200 | 0.8317 | 0.8374 | 0.8312 | 0.7825 | 0.8484 |
| **Statistical Baselines** | | | | | | |
| AutoETS | 0.9000 | 0.8939 | 0.9081 | 0.8688 | 0.8566 | 0.9306 |
| Seasonal Naive | 1.0000 | 1.0000 | 1.0000 | 1.0000 | 1.0000 | 1.0000 |

