# OpenReview forum: "Air Quality Arena: A Large-Scale Multi-Region Ground Monitoring Dataset and Benchmark for Air Quality Forecasting with Time-Series Foundation Models"
_ICML.cc/2026/Workshop/FMSD — FMSD @ ICML 2026 Poster_

### Official Review · Reviewer_TXHg · 2026-05-20

**Rating:** 9
**Confidence:** 4

**Review:**

## Summary
The paper introduces AtmoBench, a large-scale, multi-region dataset and benchmark designed to evaluate Time Series Foundation Models (TSFMs) on short-term air quality forecasting. Spanning three years, the dataset includes over 14,000 station-pollutant series covering six major pollutants across seven countries and four continents. The authors conduct an extensive evaluation of 11 zero-shot TSFMs against 6 classical and supervised machine learning baselines. The findings demonstrate that TSFMs, particularly a cross-modal architecture (VisionTS++), consistently outperform optimized classical baselines in zero-shot settings.

## Strengths
- Exceptional Dataset Diversity: The benchmark successfully addresses the geographic and pollutant narrowness of existing datasets.
- Rigorous Evaluation Design: The evaluation uses a highly practical, asymmetric comparison: it pits zero-shot TSFMs against supervised models that were directly fitted to the target series.
- Workshop Relevance: The paper is a perfect fit for the FMSD workshop. It directly addresses the call for new benchmarks evaluating structured foundation models on time-series tasks, measuring scalability, and highlighting the capabilities of cross-modal architectures (VisionTS++).
- Reproducibility: The dataset and evaluation framework are open-sourced, and careful preprocessing steps are well-documented, enabling easy community adoption.

## Areas for Improvement
- Lack of Covariates: The models currently operate in a strictly univariate setting. Since air quality is highly dependent on meteorological conditions and emission sources, the lack of covariate data is a limitation, though the authors correctly identify this as a direction for future work.
- Difficult Pollutants: Performance on SO2 and O3 remains weak across all models, with some supervised baselines even exceeding a normalized MASE of 1.0. Expanding on why these specific pollutants exhibit such complex distributional shifts would add valuable depth to the analysis.

## Detailed Comments
- The finding that VisionTS++, which renders time series as images to leverage vision foundation models, achieved the top score is highly compelling. This responds to the FMSD workshop's interest in multimodal structured foundation models.
- It was commendable to explicitly check the TSFM pretraining corpora for data contamination, finding only negligible overlap.

## Justification of Score
According to the FMSD workshop reviewer guidelines, acceptance is based on whether claims are supported by clear evidence and whether the audience would find the work interesting. This paper excels in both criteria. The claims regarding TSFM zero-shot capabilities are backed by an exceptionally broad and rigorous empirical setup. The resulting benchmark is highly valuable to the foundation model and climate science communities.

---

### Official Review · Reviewer_zEqZ · 2026-05-22

**Rating:** 7
**Confidence:** 2

**Review:**

## Summary
The paper introduces a new set of datasets and a benchmark of air quality forecasting. They evaluate a comprehensive set of TSFMs, and also include 2 statistical and 4 ML baselines. They show that TSFMs dominate on this task, generally beating

## Strengths
- Paper is well written and has a comprehensive coverage of datasets
- Wide variety of baselines and good number of TSFMs evaluated

## Areas for Improvement
- Create a table summarizing the key information of each constituent dataset in the benchmark. Include things like source, and all the other important metadata.
- Pre-processing should __not__ include imputation. Imputation should instead be performed at the model level. During evaluation, we should not be computing performance on these imputed values as it may bias towards a certain type of model.
- Not an expert on this domain, but it seems odd that all models are operating in purely univariate mode.
- Other benchmarks seem to care about multiple horizons, but this paper only

## Justification of Score
- While paper has some drawbacks, it looks good overall with a comprehensive benchmark and wide number of datasets. The application to a particular domain is within scope of the workshop, and would interest participants of the workshop.